# Population differences of chromosome 22q11.2 duplication structure predispose differentially to microdeletion and inversion

David Porubsky [1,2], DongAhn Yoo [1], Nidhi Koundinya [1], Erika Souche [3], Philip C. Dishuck [1], Nicolas Dierckxsens [4], William T. Harvey [1], Katherine M. Munson [1], Kendra Hoekzema[1], Daniel D. Chan[5], Tiffany Y. Leung[5], Marta S. Santos[3], Senne Meynants [3], Ann Swillen [6], Jeroen Breckpot[6], Vasiliki Tsapalou[2], Patrick Hasenfeld[2], Jan O. Korbel [2], Human Pangenome Reference Consortium*, Peter M. Lansdorp [5,7], Joris R. Vermeesch [3,6] & Evan E. Eichler [1,8] ✉

Chromosome 22q11.2 microdeletion syndrome (22q11.2DS) is mediated by high-identity polymorphic low-copy repeats (LCRA-to-D) that have been challenging to sequence characterize. We sequence-resolved 135 chromosome 22q11.2 haplotypes from diverse humans and define 63 distinct structural configurations differing in size by 11-fold for LCRA. This diversity is driven by a 105 kbp segmental duplication flanked by 25 kbp inverted repeats that arose in the apes but expanded in humans ~1 million years ago. African LCRA haplotypes are significantly longer ($p = 0.0047$) and predicted to be more protective against 22q11.2DS ($p = 1.14 \times 10^{-6}$) due to enrichment of inverted 105 kbp repeats. We identify nine distinct (including five recurrent) inversions spanning LCRA-D. Sequencing four families indicates LCRA-D deletions map to 105 kbp repeats, whereas inversions map to the 25 kbp repeats. Here, we show specific haplotype LCR architectures and recurrent large-scale inversions modulate susceptibility to 22q11.2DS and help explain its reduced prevalence among individuals of African ancestry.

The chromosome 22q11.2 region is a structurally complex region of the human genome characterized by the presence of large stretches of highly identical segmental duplications (SDs) also known as low copy repeats (LCRs). We use both terms interchangeably throughout the manuscript. These are organized into eight SD blocks labelled LCRA-H[1] that predispose this region to recurrent rearrangements among various pairs of SDs[2]. These homologous regions can misalign during meiosis via a mechanism known as non-allelic homologous recombination (NAHR) resulting in genomic rearrangements, including deletions, duplications, and inversions[3]. In ~85% of cases, NAHR occurs between LCRA and D, resulting in a deletion of a ~3 Mbp long critical region[4]. This deletion is the cause of DiGeorge or velocardiofacial syndrome, which is collectively referred to as the chromosome 22q11.2 deletion syndrome (22q11.2DS)[5]. It is recognized as the most common genomic disorder in humans with a prevalence of 1 in ~3000 live births[6]. Patients may manifest with a wide range of phenotypes, including developmental delay, intellectual disability, congenital heart defects,

[1]Department of Genome Sciences, University of Washington School of Medicine, Seattle, WA, USA. [2]European Molecular Biology Laboratory (EMBL), Genome Biology Unit, Heidelberg, Germany. [3]Laboratory of Cytogenetics and Genome Research, Centre for Human Genetics, KU Leuven, Leuven, Belgium. [4]Genomics and Regulatory Systems Unit & Marine Climate Change Unit, Okinawa Institute of Science and Technology Graduate University, Okinawa, Japan. [5]Terry Fox Laboratory, BC Cancer Research Institute, Vancouver, BC, Canada. [6]Department of Human Genetics, Centre for Human Genetics, University Hospitals Leuven, Leuven, Belgium. [7]Department of Medical Genetics, University of British Columbia, Vancouver, BC, Canada. [8]Howard Hughes Medical Institute, University of Washington, Seattle, WA, USA.*A list of authors and their affiliations appears at the end of the paper. ✉e-mail: ee3@uw.edu

and cleft-lip/palate. The deletion is also known to be a major genetic risk factor for schizophrenia[2,7,8].

Despite extensive research on the genetic and clinical aspects of 22q11.2DS, it is unknown why this syndrome occurs an order of magnitude more frequently than most other genomic disorders mediated by NAHR[6]. The structural complexity of the LCRA-D duplication blocks at the 22q11.2 locus and the resulting challenges in accurately and completely assembling this region have limited our ability to fully characterize the mechanisms driving these rearrangements. For instance, short-read sequencing methods are optimal for detecting large-scale deletions at 22q11.2; however, they are unable to distinguish the copy number of SDs located at the deletion breakpoints as unequal crossover typically occurs in stretches of paralogy with near-perfect sequence identity[9] (Supplementary Fig. 1). Previous studies using fluorescence in situ hybridization (FISH) and optical mapping[10–12] provided valuable insights into the structural diversity of this locus, but these methods lack the resolution necessary to precisely map deletion breakpoints or explore the fine-scale diversity of the underlying LCRs flanking this region. More recently, optical mapping methods were coupled with ultra-long Oxford Nanopore Technologies (UL-ONT) sequencing approaches to locally assemble and characterize breakpoints in 22q11.2DS families[13–15]. While this methodology helped refine recombinant breakpoint regions, standard assembly methods failed to fully resolve the chromosome 22q11.2 region or the complexity of underlying rearrangements in either patients or population controls.

Recent advances in long-read sequencing (LRS) technologies and assembly algorithms have addressed many of the previous limitations. Combining UL-ONT sequencing with high-fidelity (HiFi) PacBio, for example, was critical for the production of the first gapless, telomere-to-telomere (T2T) human genome reference T2T-CHM13[16], including resolution of all LCR regions[17]. Lessons learned from that effort have now been applied to develop new assembly algorithms[18,19], which have facilitated population-level sequencing of nearly complete T2T human genomes[20–22].

In this work, we present a detailed analysis of complete chromosome 22q11.2 from 135 diverse human haplotypes and six nonhuman primate (NHP) species[23], including macaque[24]. We use these data to not only understand the full extent of human genetic diversity of this complex region at the base-pair level but to put this variation into an evolutionary context revealing human-specific and population-enriched features that drive recurrent rearrangement of 22q11.2DS. We also apply these T2T sequencing technologies to the study of chromosome 22q11.2DS patients and their families identifying distinct hotspots of recurrent inversion and microdeletion. These findings help us to pinpoint predisposed haplotypes and those protected from the disease suggesting that not all parental haplotypes nor populations have equal probability of rearrangement and disease.

## Results

### Complete assembly of chromosome 22q11.2 haplotypes

We focused our analysis on a 5 Mbp region corresponding to human chromosome 22q11.2 characterized by large SD blocks denoted LCRA to LCRD with A and D being the longest and most identical to each other[2] (Supplementary Fig. 2). This region harbors ~54 protein-coding genes (Fig. 1a) and was only recently fully assembled as part of the T2T-CHM13 reference (region of interest (ROI) coordinates: 18–23 Mbp). The prior GRCh38 reference contained three large gaps in LCRA as well as two unassigned contigs that have been subsequently integrated into T2T-CHM13 (Fig. 1b). To establish a human population diversity panel

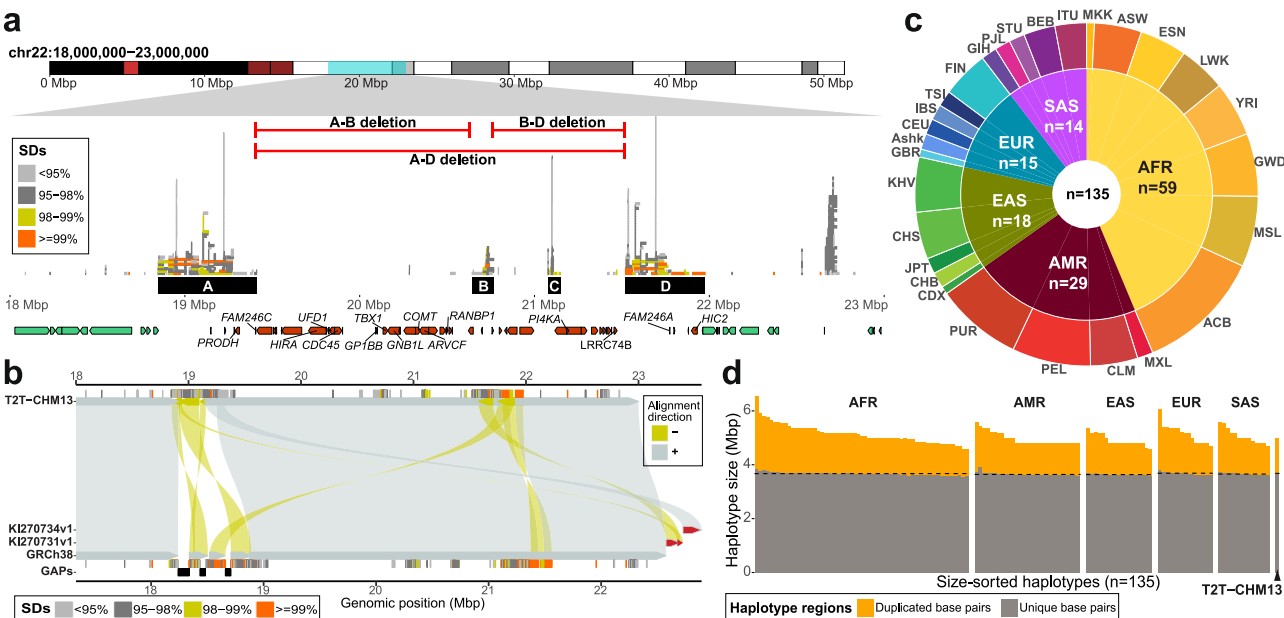

**Fig. 1 | Assembly and annotation of human chromosome 22q11.2 region.**
**a** Chromosome 22 ideogram (top) annotated with the critical region (chr22:18000000–23000000) (blue), pathogenic copy number variants (red horizontal lines), LCRA-D (black rectangles), segmental duplications (SDs; stacked horizontal bars colored by % identity in middle), and a subset of protein-coding genes (bottom, red and green arrow bars). There are a total of 76 protein-coding genes mapping between 18-23 Mbp of T2T-CHM13 reference of which 54 map between LCRA-D (in red). **b** Comparison of the GRCh38 reference (query sequence, bottom) against the T2T-CHM13 reference (target sequence, top) over 22q11.2 with direct ('+', forward - gray) and inverted ('-', reverse - yellow) alignments distinguished by color and their respective orientation. Both query and target sequences depict SD annotation colored by sequence identity and gaps in GRCh38 (black rectangles). Unassigned contigs (GRCh38) are highlighted in red. **c** A two-layer donut plot shows the counts of complete assemblies at 22q11.2 colored by major ancestries (AFR - African, AMR - American, EAS - East Asian, EUR - European, SAS - Southeast Asian). Specific 1000 Genomes Project (1KGP) populations (outer layer) are indicated by an acronym (https://www.internationalgenome.org/data-portal/population). **d** A barplot showing length variation of SD regions (orange) versus unique regions (gray); haplotypes are organized by five major ancestries. The median value of "unique" base pairs is highlighted by a dashed line per ancestry. T2T-CHM13 proportions of 'duplicated' and 'unique' base pairs are shown separately.

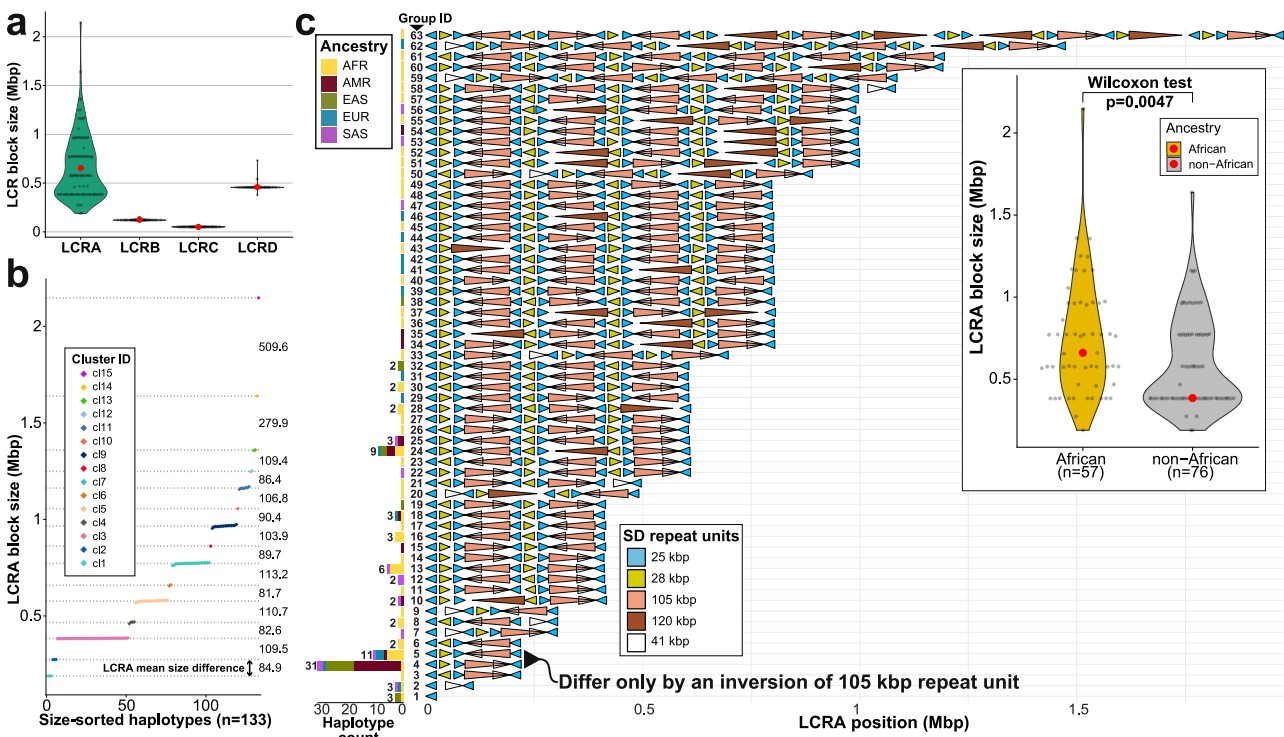

**Fig. 2 | Chromosome 22q11.2 structural diversity and duplicon higher-order structure. a** Violin plot showing length distribution for LCRA-D along with mean LCR block size (red dots). **b** Size-sorted distribution of LCRA size (colored by cluster ID) with the mean size of each cluster (horizontal dotted line) and the mean size difference between each step indicated by the number. **c** Higher-order repeat (HOR) structure of 63 distinct LCRA haplogroups (colored arrowheads show size and orientation of five repeat units 25, 28, 41, 105, and 120 kbp in length). Note that the 41 kbp repeat unit maps within the 105 kbp repeat unit and is only rarely seen separately. Stacked barplot shows the frequency of each haplogroup among continental population groups. Inset: Shows a significant size difference (Wilcoxon rank-sum test, two-sided, $p = 0.0047$) of LCRA among haplotypes of African and non-African ancestry.

of chromosome 22q11.2 haplotypes, we evaluated the completeness of 110 phased genome assemblies (220 haplotypes) generated as part of the Human Pangenome Reference Consortium release 1 (HPRC1)[21] and Human Genome Structural Variation Consortium (HGSVC)[22], including two samples with reported large-scale inversions at 22q11.2. Of these, 135 haplotypes were unrelated, assembled in a single continuous contig, and passed our assembly QC[25] (Supplementary Note 1, and Supplementary Data 1 and 2, "Methods"). Among those, there were 78 diverse individuals of which 57 have both maternal and paternal haplotypes fully assembled. The set included representation of all human ancestries (AFR - African, AMR - American, EAS - East Asian, SAS - Southeast Asian, and EUR - European) with comparable numbers of haplotypes of African ($n = 59$, 43.7%) and non-African ancestry ($n = 76$, 57.3%) (Fig. 1c, and Supplementary Fig. 3). The total length of the region varied considerably with a relatively constant 3.66 Mbp of unique sequence and highly variable number of duplicated bases (988,335–2,687,425 bp) mapping to the LCR regions (Fig. 1d, "Methods"). We observed one outlier (HG00731-H1) with ~3.91 Mbp of "unique" sequence. The subsequent analysis showed a ~270 kbp tandem duplication of that unique sequence located distal to LCRD, likely as a result of NAHR. This duplication was previously observed at a frequency of 0.002% among both, cases with neurodevelopmental delay and unaffected individuals[26,27] (Supplementary Fig. 4).

## SD population variation and higher-order structure

We assigned SDs to LCR regions A-D for 133 haplotypes based on the canonical T2T-CHM13 structure (Fig. 1)—we excluded two haplotypes that carried a large inversion which complicated such assignment. Consistent with earlier optical mapping data[28], LCRA is, by far, the most variable (>90% of total size variability), followed by LCRD in contrast to LCRB and C, which are essentially invariant (Fig. 2a, and

Supplementary Fig. 5). The median size of LCRA is 577.15 kbp with haplotypes differing by more than 11-fold in length (189.01 to 2147.85 kbp) characteristic of stepwise increase in length (Fig. 2b, "Methods"). We initially decomposed LCRA into evolutionary ancestral units, termed duplicons, using DupMasker[29], observing a set of recurrent repetitive structures in both LCRA and D (Supplementary Fig. 6). Instead of the previously predicted ~160 kbp higher-order repeat (HOR) structure[11,14], we refined the basic repeat unit to be of 155 kbp in length, consisting of a 105 kbp core flanked by 25 kbp long inverted repeats, (Supplementary Fig. 7), with base-pair resolution as opposed to the former prediction by optical mapping[28]. In addition to these repeats, we also define three other common repeats between LCRA and D of 28, 41, and 120 kbp in length, respectively, which adds up to a HOR structure composed of five common repeats (Fig. 2c, and Supplementary Fig. 8, "Methods"). This HOR structure can also be defined based on a set of repeating duplicon units (Supplementary Fig. 9, "Methods") and was generally observed by fiber-FISH, albeit not at base-pair resolution[28]. At the boundaries of 25 and 28-kbp-long HORs, we locate the previously defined pockets of palindromic AT-rich repeats (PATRRs)[30–32]. We define a common, non-AT-rich sequence motif overrepresented within PATRRs to help us determine the orientation of each AT-rich unit (Supplementary Fig. 10a, "Methods"). We find this motif present not only at the chromosome 22q11.2 region but also at other acrocentric short arms of chromosomes 13, 14 and 21, which are frequently associated with Robertsonian translocations[33] (Supplementary Fig. 10b).

For LCRA, we identify 63 structurally distinct haplogroups of various size and structural complexity (Fig. 2c, "Methods"). There are nine major LCRA haplogroups (observed ≥3 times) representing about half (54.1%, 72 out of 133) of all haplotypes while 47 haplogroups occur as singletons. We find haplotypes of African ancestry are significantly

longer than those of non-African ancestry (Wilcoxon rank-sum test, two-sided, $p = 0.0047$) (Fig. 2c, inset). The two most abundant haplotype groups (4 and 5) are comparatively short for LCRA (median: 383.3 kbp) but population stratified. Haplogroup 4, for example, is composed almost exclusively of haplotypes of non-African ancestry while haplogroup 5 is enriched among Africans and they differ only by an inversion of the 105 kbp repeat unit (Fig. 2c). In contrast to LCRA, distal LCRD shows far fewer structural haplogroups ($n = 9$) with two major ones representing 93.23% (124 out of 133) of all haplotypes. These two haplogroups (2 and 3) differ only by a single inversion (~48.65 kbp in size, flanked by ~70 kbp SDs) at the distal portion of LCRD. In contrast to LCRA, the most common LCRD haplotypes are distributed similarly among African and non-African ancestries although all remaining unique haplotypes ($n = 9$) are of African ancestry consistent with greater genetic diversity (Supplementary Fig. 6).

Parsimoniously, we explain most of the stepwise increase in LCRA size as a result of the cumulative addition of the basic HOR units (155 kbp + 28 kbp) flanked by PATRRs (~10 kbp) via NAHR likely occurring between directly oriented 25 kbp repeats. This results in a unit increase of ~193 kbp, distinguishing the most abundant LCRA haplotype lengths (Supplementary Fig. 11). The net effect is that the PATRRs and HOR units create a set of embedded palindromes that predispose LCRA to recurrent insertions, deletions, and inversions among directly and indirectly oriented repeats (Supplementary Figs. 12 and 13). When assessing the sequence identity between all paralogous copies of defined HORs, we find most are 99% identical with one exception—the 25 kbp repeat shows significantly greater divergence (Supplementary Fig. 14). We tracked the reduced sequence identity of the 25 kbp unit to a GC-rich 27 bp variable number tandem repeat (VNTR) that differs in length among paralogs and alleles and is transcribed as part of an exon of a long noncoding RNA (*FAM230*) (Supplementary Figs. 15 and 16). We provide a detailed gene annotation summary within the 22q11.2DS region, including the flanking LCRA-D regions based on mapping of long-transcriptomic Iso-Seq data (Supplementary Note 2).

## Evolutionary history of the chromosome 22q11.2 region

In order to reconstruct the evolutionary history of the chromosome 22q11.2 in humans, we compared human haplotypes to the recently completed NHP genomes (chimpanzee - PTR, bonobo - PPA, gorilla - GGO, and Bornean (PPY) and Sumatran (PAB) orangutans)[23] (Fig. 3a, b). Overall, the majority (95.49% or 127/133) of human chromosome 22q11.2 haplotypes are expanded >2-fold for SD content (range: 313–2107.8 kbp; median: 687 kbp) when compared to the syntenic regions among other great apes (range: 103–464.9 kbp; median: 268.4 kbp) due to an increase in copy number of the HOR units (Fig. 3b). Comparing the chimpanzee haplotype to the most similar human (HG02018, EAS from Vietnam) haplotype, we observe that the proximal 25 kbp and large part (~97 kbp) of the 105 kbp repeat unit map to LCRC whereas the 105 and 25 kbp repeat units typically reside in both LCRA and D in humans. The distal 25 kbp repeat and the remaining 8 kbp of the 105 kbp repeat unit map to LCRD in chimpanzee. This suggests a transposition from LCRC into D during human evolution to constitute the full 155 kbp HOR structure as well as its association with PATRRs (Fig. 3c). These data further support the spread and expansion of the 105 kbp repeat unit from LCRD into LCRA where it appears in multiple copies (1–6 copies) while in LCRD it is conserved as a single unit (Supplementary Fig. 17). Based on the comparison to chimpanzee, we observe three ancestral human LCRA structures followed by three intermediate structures supporting stepwise addition of repeat units at LCRA (Supplementary Fig. 18).

We tracked the phylogeny and synteny across primates for each of the 25 kbp, 28 kbp, 105 kbp, and 120 kbp SD repeat units. We identified the underlying smaller duplicons independently (marked i-v in Fig. 3d and vi-viii in Supplementary Fig. 19) based on synteny among the apes using the macaque genome as an outgroup[24]. While duplicons i, ii, and

iv originate as singleton or doubleton syntenically on chromosome 22 (Fig. 3a), others such as duplicon iii map to a single copy on chromosome 1 (human synteny) and, thus, are predicted to have duplicatively transposed interchromosomally from chromosome 1 to chromosome 22 in the common ancestor of human, chimpanzee, and gorilla. Unlike other duplicons, we find that duplicon v is relatively young and shared between human and *Pan* lineages, but is missing from gorilla and more diverged genomes. Tracing the source sequence of duplicon v in macaque, we find more than half of the sequence was untraceable while the remaining half is composed of three ~4 kbp units, all of which map to different chromosomes (chr8, chr10 and chr15). Comparing primate genomes, we observe a gradual increase in higher-order duplication complexity as we span the primate tree from macaque, gorilla, *Pan* and then to human with the data suggesting that the higher-order structure arose by a series of events (translocation, transposition, and juxtaposition of the smaller duplicon units as illustrated in Fig. 3d, and Supplementary Fig. 20, "Methods"). Notably, we also find that more than 94% of duplication blocks present at chromosome 22q11.2 demarcate the boundaries of inversions between humans and NHPs, suggesting the formation of these higher-order structures was frequently accompanied by inversion toggling[34] of the unique sequence bracketed by SDs (Supplementary Fig. 20).

Wherever possible, we constructed a phylogenetic tree for each of the smaller duplicons, including duplicon iii, to estimate the time when more complex SD structures formed (Fig. 3e, and Supplementary Figs. 21–31). Calibrating based on macaque–human divergence time of 28.8 million years ago (MYA), we estimate, for example, that the duplicative transposition (duplicon iii) from chr1 to chr22 along with the juxtaposition of i-iii duplicons occurred 14.5 [12.7–16.0] MYA consistent with the comparative analysis (Fig. 3e, and Supplementary Fig. 20). The segment further fused with duplicon iv ~7.3 [6.0–9.5] MYA giving rise to the ~130 kbp structure present in both human and *Pan* lineages. The complete higher-order structure arose specifically in the human lineage and we estimate the 25 kbp repeat unit began to duplicate 2.1 [1.7–2.5] MYA—two of three human haplotypes with LCRA structures most similar to chimpanzee (HG00621-H2 - EAS from China and HG02018-H1 - EAS from Vietnam discussed above) form a distinct outgroup when compared to all human copies (Supplementary Figs. 18 and 20). We estimate that LCRA and D diverged even more recently, 1.0 [0.8–1.2] MYA with the majority of subsequent expansions and conversions occurring during the emergence of *Homo sapiens* for LCRA (coalescent time of 0.7 [0.5–0.9] and 0.8 [0.7–1.0] MYA) and LCRD (coalescent time of 0.6 [0.4–0.8] MYA) (Fig. 3e).

## Population stratification of chromosome 22q11.2 inversions

The 22q11.2 region has been subjected to a large number of structural rearrangements, especially large inversions distinguishing humans and NHPs (Supplementary Fig. 31). We specifically focused on the identification of large-scale inversion polymorphisms between LCRA-D in humans whose existence has been controversial due to technological challenges in their detection[35]. We also considered inverted duplications that change the orientation of HOR units within the LCRs but do not affect the flanking sequence. Among available genome assemblies from the HPRC2 (release 2; $n = 232$) and HGSVC ($n = 65$) as well as pooled Strand-seq data[34] ($n = 279$), we identified 11 distinct large-scale inversions (mean size 1.9 Mbp) (Supplementary Data 3, and Supplementary Fig. 32, "Data availability"). These include seven carriers of the largest (~2.28 Mbp) A-D inversion extending across the chromosome 22q11.2DS critical region. Based on haplotype analysis, five of these are predicted to be, in fact, independent, recurrent mutations and, thus, each large inversion is extremely rare, occurring at a frequency <1% in the population (Fig. 4a, and Supplementary Fig. 33). We also observe inversions of LCRA-B (1.7 Mbp, HG01695-EUR, HG01175-AMR), LCRC-D (0.44 Mbp, HG03139-AFR), and LCRB-D (0.99 Mbp, HG03471-AFR)—all of which were resolved at the sequence level (Fig. 4a, and

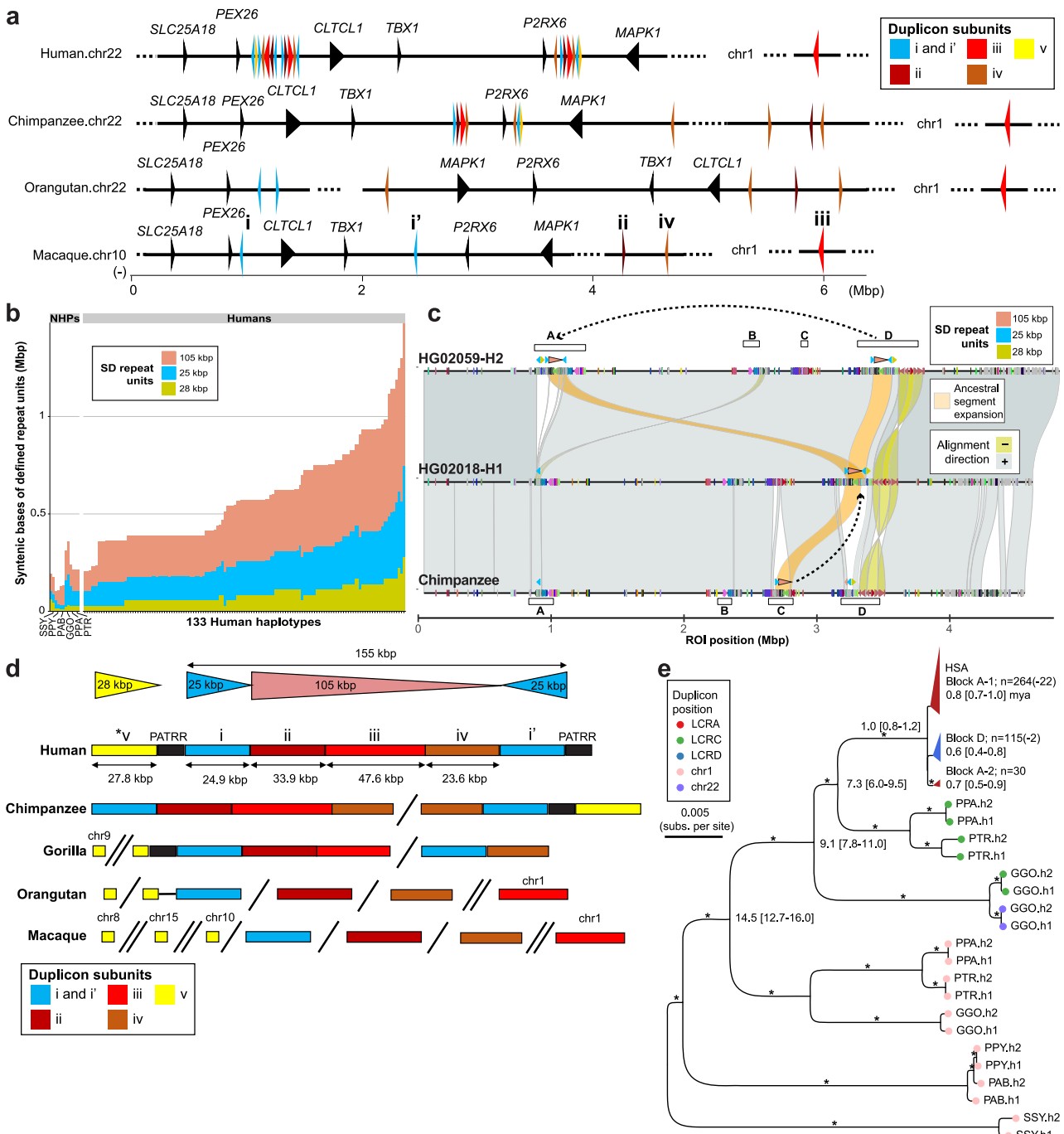

**Fig. 3 | Evolutionary history of 22q11.2 and emergence of higher-order SD structure. a** Origin of individual duplicon subunits coming from multiple ancestral loci in macaque and great ape (Bornean orangutan, chimpanzee, and human) genomes. The duplicons are indicated by different colors with an arrowhead indicating the direction: blue - duplicon i, brown - ii, red - iii, and orange - iv. Syntenic genes are indicated as black arrowheads with gene symbols above. **b** Stacked barplot showing amount of base pairs for the 25 kbp (blue), 28 kbp (yellow), and 105 kbp (light red) repeat units in humans and nonhuman primates (NHPs). **c** Miropeats-style plot showing alignments (direct - gray, '+' and inverted - yellow, '-') between chimpanzee (AG18354_PTR-H1, bottom) and two human samples (HG02018-H1, middle, and HG02059-H2, top) at 22q11.2. The DupMasker annotation for the query and target sequence are shown as directional arrowheads colored by duplicon ID. Alignments of 105 and 25 kbp long repeat units between chimpanzee and human haplotypes are highlighted by orange color (aligned region ~130 kbp). Positions of LCRA-D in both target and query are shown as empty rectangles.

A dashed arrow shows transposition of the chimpanzee repeat unit from LCRC to D and then expansion from LCRD to A in humans. **d** Inferred structure of the complex duplication unit composed of the five smaller duplicons. From top to bottom, the largest homologous units located in chr22 of human to macaque are shown, which differ in size. The full length *v duplicon is absent in gorilla, orangutan, and macaque but exists in fragments of 4 kbp or smaller. **e** Maximum likelihood phylogenetic tree of subunit iii (*n* = 421, outgrouped with macaque), based on 27.8 kbp of conserved sequence. The human copies include 274 and 135 sequences originating from LCRA and D, respectively. The number of sequences corresponding to each group is indicated; the numbers in parentheses represent non-A or -D block copies within each clade. Ortholog copies are from 12 haplotype assemblies of chimpanzee (PTR), bonobo (PPA), gorilla (GGO), Bornean orangutan (PPY), Sumatran orangutan (PAB), and siamang (SSY). Bootstrap support >95 is indicated by*.

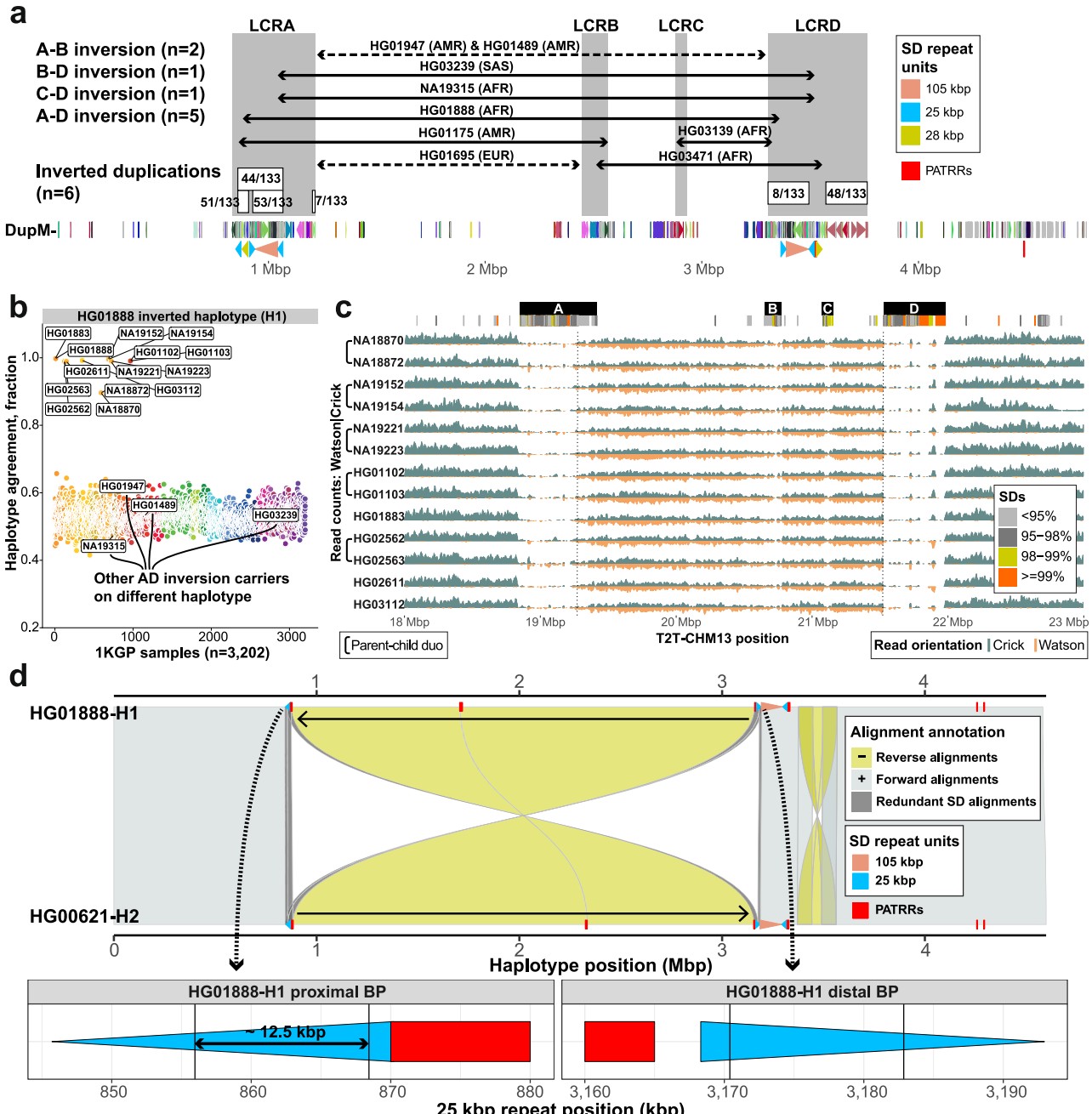

**Fig. 4 | Chromosome 22q11.2 human inversion polymorphisms and breakpoints. a** Overview of all inverted ranges and their frequency with respect to a simple 22q11.2 haplotype (HG01890-H1). Gray rectangles highlight LCRA-D positions. Solid black arrows mark assembly-resolved inversions; dashed black arrows mark Strand-seq-supported inversions. Each range is marked by a sample that carries the inversion. Inverted duplications found within LCRA and D are marked by white rectangles with a number denoting the inversion frequency. Below is a DupMasker (DupM) annotation shown as directional arrowheads colored by a duplicon ID. Further below are SD repeat units (25, 28, and 105 kbp) shown as colored arrowheads. PATRR positions shown as red rectangles. **b** Predicted carriers of inverted haplotype HG01888-H1 among 1KGP sample panel ($n = 3202$)[38]. Each dot represents a single sample colored by a population ID for the five major ancestries (AFR - African, AMR - American, EAS - East Asian, SAS - Southeast Asian, EUR - European). Samples labelled with their IDs ($n = 13 + $ HG01888) are predicted carriers of an inverted haplotype (HG01888-H1) based on haplotype concordance. **c** Read-

coverage profiles of Strand-seq data over the 22q11.2 region summarized as binned (bin size: 10 kbp, step size: 1 kbp) read counts represented as bars above (teal; Crick read counts) and below (orange; Watson read counts) the midline. Dotted lines highlight the inversion region between LCRA and LCRD. Equal coverage of Watson and Crick counts represents a heterozygous inversion as only one homolog is inverted with respect to the reference (T2T-CHM13). Above the SD annotation is colored by sequence identity with marked boundaries of LCRA-D as black rectangles. **d** Miropeats-style plot showing the alignments (direct '+' - gray, inverted '-' - yellow) between direct (HG00621-H2) and inverted (HG01888-H1) haplotypes. Positions of 105 (red) and 25 (blue) kbp SD repeat units are shown on top of the query and target sequence along with PATRRs shown as red rectangles. Redundant alignments present at the inversion boundaries are highlighted in dark gray. Below there is a visualization of predicted inversion breakpoints (-12.5 kbp region) within inversely oriented 25 kbp repeat units in proximal (LCRA) and distal (LCRD) regions.

Supplementary Fig. 34). Of note, these large-scale inversions are restricted to African and admixed American populations, and we observe a significant enrichment among African Americans (Fisher's exact test, two-sided; $p = 0.02883$, odds ratio: 5.232496) (Supplementary Data 3, and Supplementary Fig. 35). With respect to inverted duplications, we identified six commonly inverted sites (mean size 131.5 kbp) within LCRs of which four reside within LCRA and two within LCRD (Supplementary Fig. 36a–c, and Supplementary Data 4, "Methods"). These smaller inverted duplications range in length from ~12 kbp to up ~208 kbp but are much more common (33.1% to 39.8% allele frequency among 133 haplotypes) when compared to the large inversions (Supplementary Fig. 36d). There was one exception, however, which was a proximal inversion mapping to LCRD present in only eight haplotypes (6% allele frequency—all of African ancestry).

We selected one of the large A-D inversions (HG01888-H1) originally discovered using Strand-seq data[36] and subsequently assembled in this study (Fig. 4a) for population genotyping against the full 3202 from 1000 Genomes Project (1KGP) diversity panel[37,38]. Using simple genotyping concordance for single-nucleotide polymorphisms (SNPs) identified on this haplotype, we predicted 13 samples with this inverted haplotype (Fig. 4b, "Methods"). We tested for the presence of the inversion experimentally using Strand-seq, confirming all 13 samples as bona fide heterozygous carriers of this large inversion (Fig. 4c). Among the 13, five correspond to parent–child duos, confirming germline transmission of the event (as opposed to potential somatic rearrangements). Thus, we identified eight unrelated carriers for an estimated worldwide allele frequency of 0.16% (8 out of 5008 haplotypes from 2504 unrelated individuals) of this particular inverted haplotype. There was, however, almost complete stratification of this inversion between African and non-African samples with 7/8 being identified from 1KGP samples collected from the African continent (Fig. 4b).

The inverted haplotype in a single non-African sample was of admixed ancestry (PUR-HG01102/3) and we determined that this segment is ancestrally of African origin (Supplementary Fig. 35). We repeated the genotyping for the remaining four LCRA-D inversions, but each was predicted to occur as a single occurrence as opposed to multiple carriers of HG01888 inversion (Supplementary Fig. 37). Similarly, we predict other large inversions (A-B, B-D and C-D) as a single occurrence, except for the A-B inversion in HG01175 predicted to have one additional carrier (HG01074) (Supplementary Fig. 38).

In order to map the breakpoints of the inversions and inverted duplications, we characterized the selected completed assemblies ("Methods"). In cases such as HG01888-H1 and NA19315-H2, it was necessary to generate additional LRS data to fully assemble the region (Fig. 4a). Detailed analysis of HG01888-H1, for example, refines the breakpoint to a ~12.5 kbp segment mapping within inversely oriented 25 kbp repeat units (Fig. 4d, and Supplementary Fig. 39, "Methods"). Similarly, we mapped A-to-D inversion breakpoints in samples NA19315 and HG03239. Despite these inverted haplotypes being more complex than in HG01888, we narrowed down the inversion breakpoint to the vicinity of the 25 kbp repeat unit (Supplementary Fig. 40). Characterizing the inversion that occurred between LCRB and D in sample HG03471 predicts that the breakpoint likely occurs between PATRRs as there is no copy of 25 kbp repeat unit in LCRB (Supplementary Fig. 34). For inverted duplications we found all but one of the LCRA inverted duplications ($n = 3$) flanked by 25 kbp long repeats. In contrast, proximal and distal inversions in LCRD are mediated by longer 40.7 and 71 kbp long inverted repeats, respectively (Supplementary Fig. 36).

## Predisposition to 22q11.2DS predicted to differ by ancestry

Because the majority of chromosome 22q11.2 recurrent microdeletions arise as a consequence of NAHR between LCRA and LCRD driven by the 105 kbp SD repeat unit[14], we examined the orientation and percent identity of this repeat unit among 133 sequence-resolved haplotypes (Fig. 5a). We find that about a third of all haplotypes

($n = 43$) carry exactly one copy of the 105 kbp repeat unit in both LCRA and D, differing only by an inversion of the 105 kbp repeat unit in LCRA. This renders the 105 kbp repeat units in either a direct ($n = 31$) or an indirect ($n = 12$) orientation between LCRA and D (Supplementary Fig. 41) making them, theoretically, either predisposed or protected for A-to-D deletion, respectively. We extended this analysis to all 133 human haplotypes and tracked the orientation of all pairs of the 105 kbp repeat unit mapping between LCRA and D ("Methods"). We classified and rank-ordered 22q11.2 haplotypes as either predisposed to LCRA-to-D inversion or A-to-D deletion (Fig. 5b, "Methods") under the assumption that more SD pairs of directly orientated 105 kbp repeat units would favor deletion formation while those in reverse orientation would favor inversion formation[3,26]. We find significantly more haplotypes of African ancestry are predicted as being protected against 22q11.2DS than those of non-African ancestry (Fisher's exact test, two-sided, Bonferroni corrected, $p = 0.008$, odds ratio: 4.5) (Fig. 5c).

However, this predisposition analysis only considers intrachromosomal (cis) NAHR within a single human haplotype. It has been shown that NAHR at 22q11.2 arises more frequently (in about 95% cases) as interchromosomal (trans) NAHR events between maternal and paternal haplotypes during meiosis[39] (Figs. 5a and 6a). To assess this, we identified genomes where both haplotypes were fully sequenced and assembled. In total, we identified 233 samples among HGSVC and HPRC2 assemblies with both haplotypes fully assembled and compared the sequence identity and relative orientation of all possible interchromosomal pairs of the 105 kbp repeat unit between LCRA and D (Fig. 6a, "Methods", "Data availability"). We find the sequence identity of both cis and trans interactions, between the 105 kbp units within LCRA and D, to be >99% with the reverse-oriented SDs having a slightly higher sequence identity (0.1% increase) than those in forward orientation (Supplementary Fig. 42). In total, we predict 37 protected and 39 predisposed samples given the interchromosomal orientation of LCRA and D either in reverse or forward orientation, respectively (Fig. 6b, and Supplementary Data 5). Again, in line with intrachromosomal analyses, African genomes are predicted to be significantly less predisposed (Fisher's exact test, two-sided, Bonferroni corrected, $p = 1.14 \times 10^{-6}$, odds ratio = 25.8) to 22q11.2DS consistent with the >3-fold reduced prevalence reported for African Americans epidemiologically[40,41]. In addition, we predict that individuals of East Asian ancestry are significantly enriched (Fisher's exact test, two-sided, Bonferroni corrected, $p = 1.29 \times 10^{-7}$, odds ratio = 0) for predisposed haplotypes, although there is no epidemiological data at present to support this genomic prediction (Fig. 6c).

## Sequence breakpoint characterization of 22q11.2DS patients

To directly test the relationship between LCRA and D structures and sporadic chromosome 22q11.2 deletions associated with disease, we sequenced and assembled the 22q11.2 region from four families where a de novo deletion had occurred in a child with developmental disabilities. Because the transmitting parent was already known in three cases[14], we sequenced and assembled parent–child duos, while for the fourth, we considered all three members of the parent–child trio ("Data availability"). Using LRS and assembly approaches, we fully assembled and performed QC analysis on the chromosome 22q11.2 region for each transmitting parent and a child (proband) (Supplementary Note 1). We compared the phased assembly of the proband (with A-to-D deletion) with both haplotypes (H1 and H2) of the transmitting parent to define at the sequence level the breakpoint of the rearrangement. Given the accuracy of the assembly, near-perfect sequence identity could be used to readily track parental haplotypes against the child's assembled chromosome (Fig. 6d, as we demonstrated previously[42]), in order to define recombination breakpoints.

For family 1 (AD009), a single breakpoint region was readily identified, deleting ~2.5 Mbp from H1, as a result of an unequal crossover between H1 and H2 likely when the quadrivalent was formed during meiosis I, allowing for interchromosomal exchange between

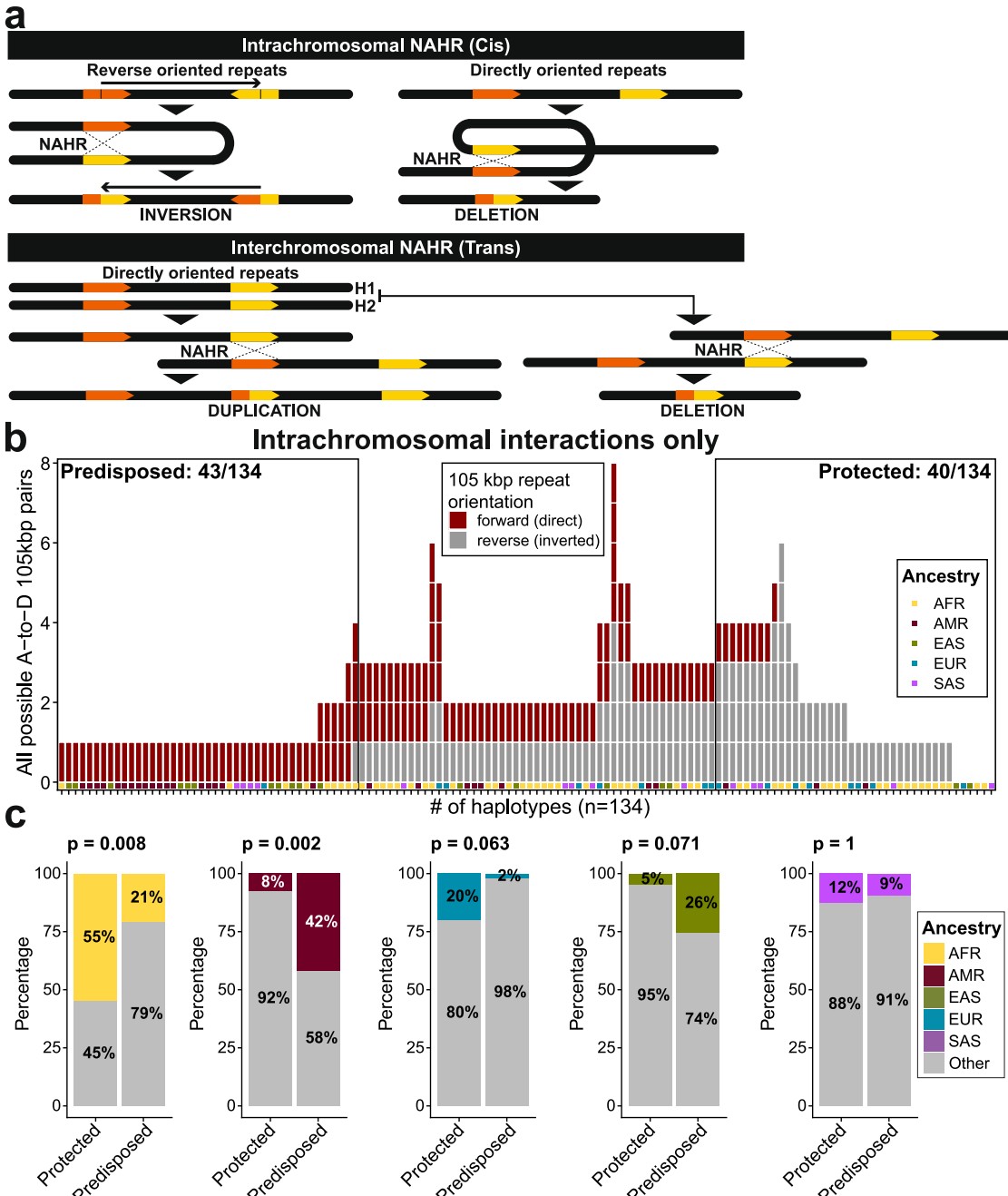

**Fig. 5 | Predisposition to 22q11.2DS via intrachromosomal NAHR. a** Model of intra- and interchromosomal non-allelic homologous recombination (NAHR) and its possible outcomes depending on the relative orientation of segmental duplications (or low copy repeats). **b** Counts of all possible LCRA and D interactions of the 105 kbp repeat unit per single haplotype ($n = 134$, including T2T-CHM13 haplotype). Only intrachromosomal (cis) interactions are evaluated. Direct (forward)-oriented pairs are dark red and inverted (reverse) pairs are gray. At the base of the plot, we show the ancestry origin of each haplotype. **c** Evaluation of the proportions of each ancestry (AFR - African, AMR - American, EUR - European, EAS - East Asian, SAS - Southeast Asian) among haplotypes defined as either predisposed or protected against 22q11.2 deletion syndrome with respect to the rest of the ancestries ('Other' - gray). We used Fisher's exact test (two-sided) to test the significance of the difference between these proportions. Above each barplot is a reported $p$-value after Bonferroni correction.

homologs. In this family, the deletion breakpoint occurs within the 105 kbp repeat unit (Fig. 6e), and we refine the breakpoint to a very narrow range of 100 bp (Fig. 6c, and Supplementary Fig. 43, "Methods") mapping nearby an exon of the transcribed *GGT2* gene family. In family 2 (AD010) and family 3 (AD013), we observe a nearly identical pattern of haplotype exchange between parental homologs as in family 1, consistent with unequal crossover within the 105 kbp repeat. Here, we narrowed down the deletion breakpoint to a 200 bp and 2303 bp region for families 2 and 3, respectively (Supplementary Fig. 43). Last,

in family 4 (the publicly available parent–child trio; "Data availability"), we again see the expected interchromosomal exchange between parental homologs at the deletion breakpoint mapped within 1150 bp range. In addition, we observe a large-scale insertion (40.8 kbp) at the deletion (Supplementary Fig. 44). Taken together, all four families have deletion breakpoints mapping to the 105 kbp SD repeat unit albeit at different locations (Fig. 6e). These breakpoints can be found within or in the vicinity of the high-identity regions (≥99.5%) clustered in two locations (around 50 kbp and 74 kbp) of the 105 kbp repeat unit

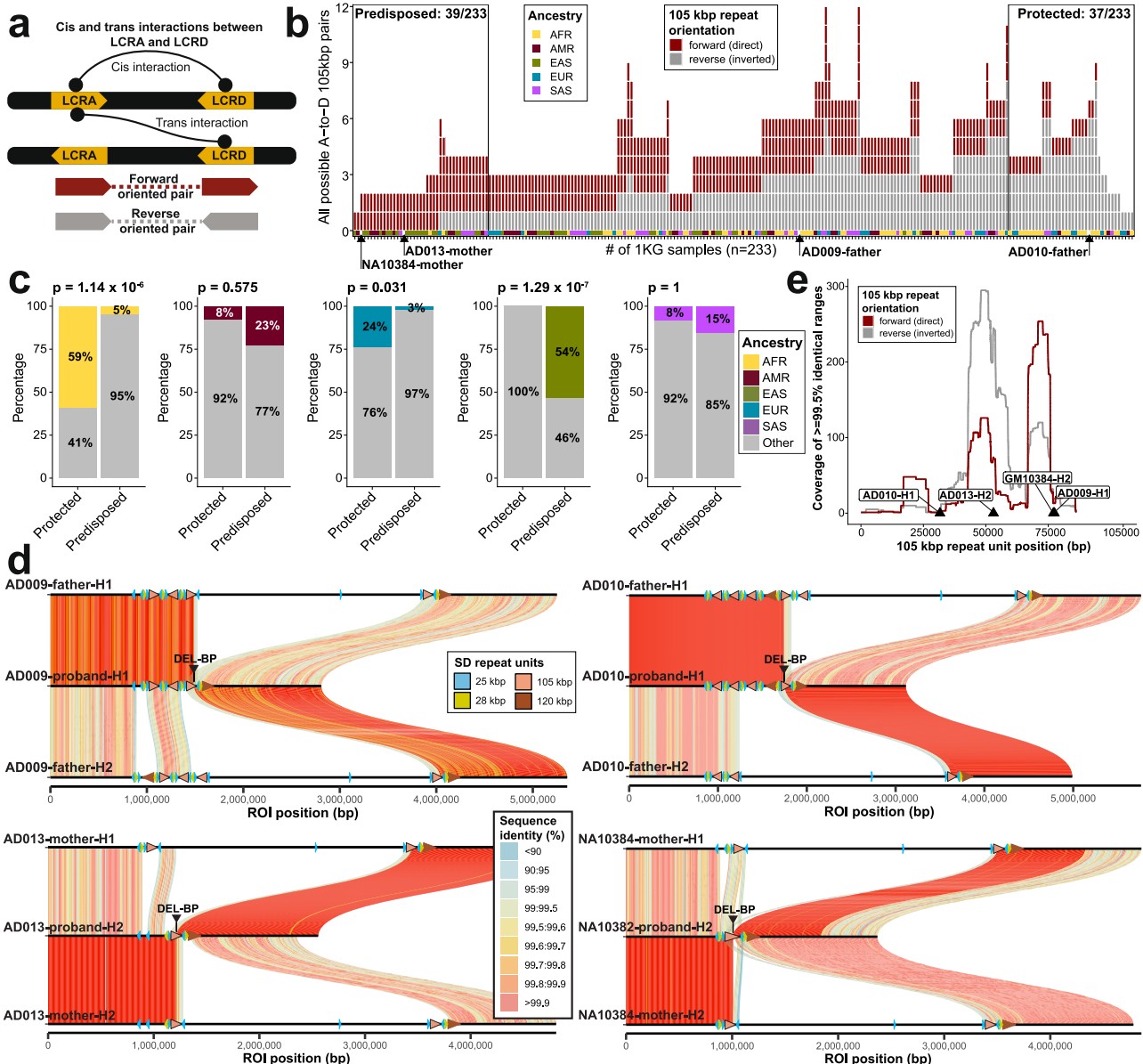

**Fig. 6 | 22q11.2DS predisposition and microdeletion rearrangements.**
**a** Pictogram of intrachromosomal (cis) and interchromosomal (trans) interactions between the 105 kbp repeat unit within LCRA and LCRD. Forward- and reverse-oriented 105 kbp repeat units between LCRA and D shown in dark red and gray, respectively. **b** Counts of all possible LCRA and D interactions of the 105 kbp repeat unit per sample ($n = 233$, plus four 22q11.2DS transmitting parents). Only interchromosomal (trans) interactions are evaluated (see Fig. 5 for intrachromosomal interactions). Direct (forward)-oriented and inverted (reverse) pairs are in dark red and gray, respectively. The base of the plot shows the ancestry origin of each haplotype. Black arrows mark the results for four transmitting parents. **c** Evaluation of the proportions of each ancestry (AFR - African, AMR - American, EUR - European, EAS - East Asian, SAS - Southeast Asian) among haplotypes defined as either predisposed ($n = 39$) or protected ($n = 37$) against 22q11.2DS with respect to the remaining ancestries ("Other" - gray). We used Fisher's exact test (two-sided) to test the significance of the difference between these proportions. Above each barplot is

the reported $p$-value after Bonferroni correction. **d** Visualization of binned (bin size: 10 kbp) sequence identity between probands with respect to both haplotypes (H1 and H2) of a transmitting parent for four families. The proband haplotype is in the middle while parental haplotypes H1 and H2 are at the top and bottom, respectively. Alignments are colored by sequence identity and likely inherited portions of parental haplotypes in the proband are highlighted by fully colored alignments (darker shades of red) while others are shown with 50% transparency. On top of each haplotype are HOR annotations depicted as arrowheads colored by SD repeat units. Refined deletion breakpoints within the 105 kbp repeat units are marked by black arrowheads. **e** Coverage summary of the longest continuous region of high identity (≥99.5%) among all alignments between all possible interchromosomal interactions of 105 kbp SD repeat units per sample. We distinguish the coverage for the direct (forward, dark red) and inverted (reverse, gray) alignments. Putative deletion breakpoint positions in four probands denoted as black arrowheads.

(Fig. 6e, and Supplementary Data 6). While this association with the larger 105 kbp repeat unit was predicted before[15,28], we now provide fully resolved haplotypes with base-pair resolution.

Finally, we projected all haplotypes of A-D deletion transmitting parents ($n = 4$) against our population genetic analysis of fully assembled haplotypes from 233 samples where we had initially

predicted predisposition to or protection against the 22q11.2DS based on the orientation and sequence identity of interchromosomal interactions between LCRA and D (Fig. 6b). When rank ordering parental haplotypes on our scale of A-D deletion predisposition, we find two out of four parents are predicted to be fully predisposed to A-D deletion. One transmitting parent ranks as equally predisposed

and protected, while the last parent ranks as more protected, suggesting that the sequence structure offers predictive power but is not absolutely deterministic of rearrangement susceptibility as long as there are still SDs in a forward orientation that can predispose to NAHR (Fig. 6b).

## Discussion

Using T2T genomes, we provide a comprehensive view of the evolution and population diversity of one of the most structurally dynamic, disease-associated regions of the human genome with the aim of providing new mechanistic insights into its origin and mutational liability. Complete sequencing of human chromosome 22q11.2 revealed two remarkable features of human genetic diversity, namely, the rapid expansion of the higher-order structure of LCR copy number polymorphism and the low-level frequency of large-scale inversions. The discovery of large, rare, and recurrent inversions adds chromosome 22q11.2 microdeletion syndrome to the growing list of genomic disorders where inversion polymorphisms associate with hotspots of recurrent deletion and duplication[34]. The frequent evolutionary and ongoing inversions are tightly coupled to the restructuring of the SDs, although parental haplotypes do not carry or directly predispose to rearrangement in the case of 22q11.2 microdeletions[43]. Here, we sequence resolve six distinct large inversions (ranging in size from 0.44 Mbp to ~2.28 Mbp). We further validate a subset by orthogonal Strand-seq analyses of the corresponding cell lines and confirm germline transmission for eight events by examining parent–child trios (Supplementary Data 3).

The larger inversion polymorphisms are exceedingly rare (<1%) but occur recurrently. For instance, we find the A-D inversion, spanning the 22q11.2DS critical region, occurs independently in five different individuals (two African, two American, and one Southeast Asian ancestry). Remarkably, almost all large-scale inversions occur among individuals of African and admixed American ancestry. Population screening using genotype concordance of SNPs underlying the inverted haplotypes shows that one of the most common inversions, LCRA-D in sample HG01888 (haplotype 1), occurs at only 0.16% (8/5008) allele frequency and is exclusive to African and admixed American individuals from the 1KGP. In most other cases, the inversions are exclusive to individual families, suggesting that such inversions (unlike chromosome 8p23 and 17q21.31 inversions)[36,44,45] fail to reach high frequency in the population, possibly due to the action of purifying selection. These recurrent inversions affect dozens of unique genes mapping to the critical region and potentially the patterns of linkage-disequilibrium and gene expression. Although we focused on LCRA-D duplications, it is likely that this inversion toggling persists for other SD blocks along chromosome 22. For example, we document a large novel inversion in a single sample of African ancestry (NA20129) that extends from LCRD to LCRH positioned ~3.5 Mbp away (Supplementary Fig. 45).

The other remarkable feature was the polarized variation of LCRA compared to the other duplication blocks (LCRC-D). We show that "normal" human LCRA haplotypes can differ by 11-fold with respect to SD content—placing it among the most copy number variable euchromatic regions in the human genome[20,46]. Previous studies using fiber-FISH and optical mapping predicted extensive tandem variation in LCR copy number[10–13,28]. Sequence-based resolution and duplicon characterization, however, reveal a much more complex and nuanced pattern of alternating tandem and palindromic HOR structures. Consistent with recent genome-wide surveys[47], we observe some of the highest copy number (and therefore largest LCRA regions) among individuals of African descent. In our sample of 63 distinct LCRA structures, we find that 11 out of 15 of the longest LCRA blocks occur among individuals of African origin.

Importantly, these expansions do not always correlate with a predicted increased probability for NAHR. Considering percent

identity and repeat orientation between LCRA and D, an analysis of both interchromosomal and intrachromosomal configurations predicts that individuals of African ancestry are more likely to be predisposed to inversions as opposed to large-scale de novo deletions (Figs. 5c and 6c). Thus, despite the larger size of LCRA and higher copy number of the 105 kbp SD repeat unit, Africans are more likely to be fully protected against 22q11.2DS than those of non-African ancestry because of the preponderance of inversely oriented repeats between the LCR blocks. This sequence structure prediction matches closely an epidemiological observation made 20 years ago, where 22q11.2DS deletions were found to be three times less common among patients of African-American ancestry[40,41]. This reduced prevalence was subsequently shown not to be the result of phenotypic ascertainment biases and was hypothesized to be the result of differences in the LCR structures[48]. In contrast, we find that the LCR structures of individuals of East Asian ancestry are significantly (Fisher's exact test, two-sided, Bonferroni corrected, $p = 1.29 \times 10^{-7}$, odds ratio = 0) predisposed to interchromosomal NAHR due their predominant direct orientation of the 105 kbp repeat in between LCRA and D; however, we were not able to find clinical results to support this observation. Nevertheless, these findings suggest that it might be more important to screen individuals of East Asian descent (i.e., Japan) for 22q11.2DS given their particular structural configurations that predispose to NAHR (Supplementary Fig. 46).

The human-specific expansion of LCRA centers around an ~160 kbp SD block first described as a *GGT*-containing core duplicon[49] and then later as a module harboring long noncoding lncRNA, *FAM230A/B* genes[13,50]. Based on our sequence analysis, we redefine the boundaries of this repeat block into 105 kbp core flanked by two 25 kbp inverted repeats that itself largely overlap with *FAM230* lncRNA (Supplementary Fig. 15). Our complete sequencing of human genomes[16,20,21] as well as NHPs[23] confirms that this core duplicon expanded exclusively in the human lineage[51]. We estimate that this expansion occurred very recently during the evolution of the *Homo* genus 750 thousand years ago (kya; CI 1060 kya to 600 kya), spreading copies predominantly to LCRA and to a far lesser extent to LCRD. In orangutans and gorillas, this 105 kbp HOR structure does not exist; it is fragmented into at least three sections and distributed to other chromosomes with the fully constituted segment emerging only in *Homo-Pan* ancestral lineage. Among chimpanzees and bonobos, we find a single copy of the 105 kbp repeat unit mapping to LCRC suggesting that this likely represents the ancestral location that was subsequently transposed and expanded in the human lineage to LCRA and D. Given the central role this 105 kbp duplicon plays in mediating NAHR and the 22q11.2DS, it follows from the comparative analysis that the predisposition to this deletion syndrome likely has been recently acquired during human evolution.

The accurate sequence assembly afforded by LRS allowed us to characterize in detail the breakpoints of both naturally occurring inversions in the general population as well as pathogenic rearrangements associated with the most common form of the 22q11.2DS. We find that inversion and microdeletion breakpoints cluster to distinct regions in the LCR blocks—with inversion and inverted duplication breakpoints mapping to the 25 kbp SD repeat unit[52,53] while LCRA-D microdeletion patients map more broadly to the 105 kbp repeat unit. A recent study of 15 families using optical mapping and fiber-FISH confirmed that 13/15 families with LCRA-D deletions map to the ~160 kbp duplicon (termed RL-AD1), which represents one of the largest and most identical repeats between LCRA and D[14]. It is interesting that RL-AD1 was first described as being associated with most common non-Robertsonian constitutional translocation in the human genome t(11:22)[54]. These two features of instability are related as the PATRRs appear to demarcate the edges of the core duplicon potentially helping to explain its hyperexpansion in the human lineage.

Finally, the complete sequence of 22q11.2 region, along with characterization of the associated SNP haplotypes and SD structures should provide a powerful resource to further advance studies of human genetic disease. Several genome-wide association studies, for example, have focused on the identification of the genetic modifiers expressing schizophrenia in 22q11.2DS[9,55,56]. Because of the long-standing gaps in the reference genomes, the complexity of the SDs, the recurrent and independent inversion and rearrangements in the region, and hence the lack of linkage disequilibrium with flanking SNPs, those studies excluded the SDs—which are the source of most of the genetic variation in this region. Moreover, the existence of such large-scale inversions and their effect on expression, linkage disequilibrium, or phenotypes represents a new area of investigation. The sequence-resolved structural map of the 22q11.2 region we generated should provide a framework to include the SDs and inversions in future genome-wide association studies, allowing investigators to explore the contribution of this variation to 22q11.2DS phenotypic variability and neuropsychiatric phenotypes.

## Methods

### Clinical samples

Study approval for three family duos (AD009, AD010, and AD013) was obtained from the Medical Ethics Committee of the University Hospital/KU Leuven (S62997) along with approval for data sharing with the University of Washington (S67964). Written informed consent to participate in the study was obtained from all individuals and/or their guardians. Medical data are pseudonymized, and DNA sequences are coded. This means that the individual's identity is replaced by an identification code in the study. As a result, the sequence data cannot be traced back to a name or any other identifiable personal information. In the case of the family trio (mother - NA10384, father - NA10383, child - NA10382), the written informed consent was obtained as part of the sample collection for the NIGMS Human Genetic Cell Repository. To ensure the privacy of clinical patient data, access is restricted and requires a formal application via the European Genome-phenome Archive (EGA) ("Data availability").

### Generation of PacBio HiFi datasets

PacBio HiFi data were generated per manufacturer's recommendations. Briefly, high-molecular-weight (HMW) DNA was extracted from frozen blood or cultured lymphoblasts using the Monarch® HMW DNA Extraction Kit for Cells & Blood (New England Biolabs, T3050L). At all steps, quantification was performed with Qubit dsDNA HS (Thermo Fisher, Q32854), measured on DS-11 FX (Denovix), and size distribution checked using FEMTO Pulse (Agilent, M5330AA & FP-1002-0275.) HMW DNA were sheared with Megaruptor 3 (Diagenode, B06010003 & E07010003) using settings 28/31 or 28/30 (depending on original length distribution) and used to generate PacBio HiFi libraries via the SMRTbell Prep Kit 3.0 (PacBio, 102-182-700). Size selection was performed with Pippin HT using a high-pass cutoff of 12 kbp (NA19315), 15 kbp (NA10382), or 17 kbp (all others) (Sage Science, HTP0001 & HPE7510). Samples HG01888, NA10382, NA10383, and NA10384 were sequenced on the Sequel II platform on SMRT Cells 8M (PacBio, 101-389-001) using Sequel II Sequencing Chemistry 3.2 (PacBio,102-333-300) with 2-h pre-extension and 30-h movies, aiming for a minimum estimated coverage of 40× in PacBio HiFi reads, assuming a genome size of 3.1 Gbp. The remaining samples (AD009, AD010, AD013, and NA19315) were sequenced on the Revio platform on SMRT Cells 25 M. Samples AD009, AD010, and NA19315 were sequenced with Revio Chemistry V1 (PacBio, 102-817-900) with diffusion loading and 24-h movies or Adaptive Loading and 30-h movies (in case of NA19315). Sample AD013 was sequenced with Revio SPRQ chemistry (PacBio, 103-520-200) with Adaptive loading and 30-h movies. All Revio samples aimed for a minimum estimated coverage of 30× in PacBio HiFi reads, assuming a genome size of 3.1 Gbp.

### Generation of UL-ONT datasets

UL-ONT data were generated from DNA extracted from the lymphoblastoid cell lines (LCLs) using a modified phenol chloroform extraction protocol[57]. Briefly, $3-5 \times 10^7$ cells were lysed in a buffer containing 10 mM Tris-Cl (pH 8.0), 0.1 M EDTA (pH 8.0), 0.5% w/v SDS, and 20 mg/mL RNase A (Qiagen, 19101) for 1 h at 37 °C. Next, 200 µg/mL Proteinase K (Qiagen, 19131) was added, and the solution was incubated at 50 °C for 2 h. DNA was purified via two rounds of 25:24:1 phenol-chloroform-isoamyl alcohol extraction followed by ethanol precipitation. Precipitated DNA was solubilized in 10 mM Tris (pH 8.0) containing 0.02% Triton X-100 at 4 °C for 2 days. Libraries were constructed using the Ultra-Long DNA Sequencing Kit (ONT, SQK-ULK001 and SQK-ULK114) with modifications to the manufacturer's protocol. Specifically, ~40 µg of DNA was mixed with FRA enzyme and FDB buffer as described in the protocol and incubated for 5 min at RT, followed by heat-inactivation at 75 °C. RAP enzyme was mixed with the DNA solution and incubated at RT for 1 h before the clean-up step. Clean-up was performed using the Nanobind UL Library Prep Kit (Circulomics, NB-900-601-01) and eluted in 225 µL EB. Lastly, 75 µL of the library was loaded onto a primed R9.4.1 or R.10.4.1 flow cell for sequencing on the PromethION, with two nuclease washes and reloads after 24 and 48 h of sequencing.

### Generation of Strand-seq datasets

Low-coverage Strand-seq data were generated for 13 inversion candidate samples (NA18870, NA18872, NA19152, NA19154, NA19221, NA19223, HG01102, HG01103, HG01883, HG02562, HG02563, HG02611, HG03112). LCLs for these samples were obtained from Coriell Institute for Medical Research (Camden, NJ). After culture with BrdU for 24 h, formaldehyde-fixed nuclei with BrdU incorporation in the G1 cell cycle phase were enriched by FACS after staining with Hoechst 33258 and Propidium Iodide as described[58]. Approximately 80 single-cell Strand-seq libraries were made for each sample with the cellenONE X1 instrument (SCIENION US, Tempe, AZ). Single-nuclei isolation and picoliter to nanoliter reagent dispensing into the wells of a 5184-nanowell array was performed as described with slight modification[42,58]. Briefly, formaldehyde cross-linking of nuclei was reversed by heat, and structural nuclear proteins were digested by a heat-labile protease. Restriction enzymes AluI and HpyCH4V (New England Biolabs (Canada) Ltd, Whitby, ON) were used for fragmenting genomic DNA, and Hemo KlenTaq (New England Biolabs) was used for A-tailing. Fragments were ligated to forked adapters, UV-irradiated, and PCR-amplified with index primers pre-spotted into nanowells before library preparation. Strand-seq libraries were size-selected with AMPure XP beads, and 300 to 700-base-pair fragments were gel-purified prior to PE75 sequencing on the NextSeq 550 (Illumina, San Diego, CA).

### Generation of pooled Strand-seq datasets

To efficiently generate low-coverage single-cell Strand-seq libraries from hundreds of human LCLs, samples were processed in pools of 40 different human LCL samples from various human populations. The pools ($n = 8$) were made by Coriell Institute for Medical Research by culturing individual LCLs and mixing equal numbers of viable cells that were frozen in a single pool of 40 samples. Strand-seq libraries were made from single cells from such pools after 24-h culture with BrdU. After FACS enrichment of nuclei with BrdU, nanoliter library preparation was done in the same way as individual cell lines. Typically, around 2000 single-cell libraries were made from each pool of 40 samples. Strand-seq libraries were size selected with AMPure XP beads and fragments were gel purified prior to PE75 (250 to 450 bp) or PE150 (450 to 1000 bp) sequencing on the AVITI (Element Biosciences, San Diego, CA). After demultiplexing of single-cell Strand-seq library reads from individual wells and alignment to the human reference genome, libraries were re-assigned to individual samples within the

pool of 40 samples using publicly available reference haplotype SNP data.

## De novo genome assembly of selected samples

For this project, we assembled three individuals affected by the 22q11.2 A-D deletion along with a known transmitting parent (AD009, AD010, and AD013, LCLs). In addition, we assembled a family trio (mother - NA10384, father - NA10383, child - NA10382, LCLs) obtained from Coriell where the child has a de novo A-D deletion at 22q11.2. All above-mentioned samples were assembled using a combination of HiFi and UL-ONT reads using hifiasm (v0.19.5 or v0.24.0) assembler with the default parameters except for NA10382, which was assembled using UL-ONT only (hifiasm parameter: '–ont'; hifiasm v0.25.0). Previously reported LCRA-D inversion in HG01888 was assembled using HiFi reads only with hifiasm (v0.16.1)[19] assembler with the default parameters. The other LCRA-D inversion carrier (NA19315) was assembled using a combination of HiFi and UL-ONT reads using hifiasm (v0.19.5)[19] assembler with the default parameters (Supplementary Data 7 and 8).

## Targeted assembly of 22q11.2DS families

Hifiasm assemblies were validated by performing targeted assemblies of the 22q11.2 region with NOVOLoci (v0.5)[59]. We applied targeted assembly to each family with observable phasing errors (AD009 and Coriell trio). We assembled the region twice, using one proximal and one distal seed to increase confidence. As the proximal seed, we selected a 500 bp sequence from the CHM13 reference genome starting at position 18,000,000; the distal seed was a 500 bp sequence starting at position 23,800,000. These NOVOLoci assemblies were generated using only UL-ONT reads and therefore had a lower QV. Consequently, they were used exclusively to investigate suspected misassemblies in the hifiasm assemblies.

## Subsetting a de novo assembly sequence to a region of interest (ROI)

In total, we evaluated the completeness of 220 phased genome assemblies generated as part of the HPRC1 (release 1; 86 haplotypes–eight redundant haplotypes removed from a total of 94 haplotypes), HGSVC (130 haplotypes)[21,22], and four haplotypes from samples (HG01888 and HG03471) that carry inversion polymorphisms over the 22q11.2 region. Of these, 35 HPRC, 96 HGSVC, and four inverted haplotypes were unrelated and assembled in a single continuous contig, passing our assembly quality validations[25] (Supplementary Fig. 2). In order to extract FASTA sequence from a 22q11.2 region (ROI, chr22:18000000–23000000) in T2T-CHM13 coordinates, we aligned all available human assemblies from the HPRC and HGSVC to the T2T-CHM13 reference using minimap2 (v2.24)[60] with the following parameters: "-x asm20 –secondary = no -s 25000". Next, we used rustybam (v0.1.33, 10.5281/zenodo.8106233) and its functionality called 'liftover' in order to subset alignments in a PAF file to a desired region. Then we used such subsetted PAF files to extract the query FASTA sequence using the R package SVbyEye (v0.99)[61] and its function "paf2FASTA". We only reported those assemblies that span the ROI in a single contig, while at the same time ends of the contig do not map further than 100 kbp from the region boundaries. To evaluate each assembled haplotype for possible misjoins or collapses, we aligned PacBio HiFi reads back to each haplotype using minimap2 (v2.26)[60] with the following parameters: `-a -I 10G -Y -x map-pb --eqx -L --cs`. Such alignments were then manually evaluated for possible collapses by projecting the most and the second most abundant base at each haplotype as reported previously for NucFreq analysis[25]. Any haplotype showing an extended region where there is an observable frequency of the second most abundant base was rejected from further analysis. In total, we selected for further analysis 135 human haplotypes (of which two carry an inversion), including

extra haplotypes of the NHPs (n = 12), de novo assemblies of clinical samples (four families), and extra assembly of A-D inversion in NA19315. Similarly, we processed HPRC2 (release 2) assemblies (n = 232) ("Data availability").

## Definition of continuous LCR blocks

To define continuous LCR regions (or SD blocks) in T2T-CHM13 (v2) coordinates, we took available SD annotation from: https://s3-us-west-2.amazonaws.com/human-pangenomics/index.html?prefix=T2T-CHM13/assemblies/annotation/. Next, we subsetted the SD to our defined ROI (see above). To get a continuous set of LCRs, we collapsed gaps between consecutive SD regions that are 25 kbp in size or shorter. Then, we took only continuous blocks that contain at least 10 SDs and are 10 kbp in size and larger (Supplementary Data 9). To define LCR boundaries per haplotype, we first aligned each haplotype to itself using minimap2 (v2.24)[60] with the following parameters: `-DP -k19 -w19 -m200 -c --eqx`. Resultant self-alignments in PAF were read using the SVbyEye (v0.99)[61] function "readPaf". Such alignments were filtered using the SVbyEye function 'filterPaf' with the following parameters: `min.mapq = 0, is.selfaln = TRUE, min.align.len = 10,000, min.selfaln.dist = 20,000`. Continuous alignments were converted into genomic ranges such that any gaps 50 kbp and shorter were collapsed to create continuous ranges. Finally, we assigned to each continuous LCR an identifier (A-E) based on the overlaps with T2T-CHM13 LCR annotation mapped to each haplotype.

## Clustering LCRA by size

We took previously defined LCRA sizes (n = 133), as described in "Definition of continuous LCR blocks", and clustered them by their size using the R package ClusterR (v1.3.2) and its function "KMeans_rcpp" into 15 clusters. The number of clusters was defined by visual observation of the LCRA size distribution. We then ran the "KMeans_rcpp" function with the following parameters: `clusters = 15, num_init = 10,000, max_iters = 1000, initializer = 'kmeans++'`.

## Defining duplicon annotation for each haplotype

We defined the duplicon structure of each human haplotype (n = 135), including haplotypes from clinical samples and NHPs, using DupMasker (v4.1.2-p1)[29]. Specifically, we used the Rhodonite pipeline, available on GitHub (https://github.com/mrvollger/Rhodonite), to run separately on each selected and validated haplotype as described in "Subsetting a de novo assembly sequence to a ROI".

## Definition of shared SD repeat units between LCRA and D

Previously, a segment of -160 kbp in size was defined as a repeating unit constituting LCRA and D[11,14]. We redefined this repeat unit using a complete sequence (T2T-CHM13 reference) of the 22q11.2 region (chr22:18000000–23000000). We aligned this sequence to itself using minimap2 (v2.24)[60] using the following parameters: `-x asm20 -c --eqx -DP -m200`. We then extracted the sequence defined by the self-alignment boundaries (three copies in T2T-CHM13) and constructed the multiple sequence alignment (MSA) using MAFFT (v7.525) with parameter `--auto`. Starting from the MSA, we defined the longest continuous alignment with gaps ≤500 bp (Supplementary Fig. 7a, b). We then collapsed this region of the MSA into a consensus sequence of -155 kbp using "ConsensusSequence" of the DECIPHER (v3.2.0)[62] R package with the following parameters: `threshold = 0.5, minInformation = 0.5, includeNonLetters = TRUE`. This was followed by self-alignments of the consensus sequence to itself as described above. With this we defined -25 kbp inverted repeats at the flanks of our consensus sequence with a unique 105 kbp region in the middle (Supplementary Fig. 7c). We report 105 kbp and 25 kbp long consensus sequences as common repeat units present in both LCRA and D. We found these sequence units to overlap well with previous definitions[11,14]

of a 160 kbp repeat while adding more resolution as to which parts are unique and which are repeated (Supplementary Fig. 7d).

In addition to the 155 kbp long repeat unit, we also observed additional frequently occurring repeat units of about 28 kbp and 41 kbp. We initially defined these SD repeats in T2T-CHM13 at positions chr22:18879899–18907599 and chr22 18936513–18976199, respectively. We mapped both sequences back to T2T-CHM13 to find all locations of each sequence using minimap2 (v2.24)[60] with the following parameters: `-x asm20 -c --eqx --secondary=yes`. We then extracted, for each repeat unit, the sequence from all locations in the T2T-CHM13 and constructed MSA using R package DECIPHER (v3.2.0)[62] with its function "AlignSeqs". Subsequently, we report the consensus sequence of ~28 kbp and ~41 kbp in size as described above. Lastly, we defined ~120 kbp repeat unit observed in five copies in sample NA19129 (haplotype 1; four copies in LCRA and one copy in LCRD). As described above, we again constructed an MSA using DECIPHER (v3.2.0, function "AlignSeqs") and exported a consensus sequence of about 120 kbp in length. See "Data availability" for access to all consensus FASTA files for all SD repeat units (25 kbp, 28 kbp, 41 kbp, 105 kbp, and 120 kbp). Having consensus repeat units for 25, 28, 41, 105, and 120 kbp long segments, we mapped them back to each 22q11.2 haplotype ($n = 135$) using minimap2 (v2.24)[60] with the following parameters: `-x asm20 -c --eqx --secondary=yes -E1,0 -p 0.75` in order to obtain their coordinates in each haplotype. Note that in the case of 120 kbp repeat unit we excluded partial alignments smaller than 100 kbp.

## Definition of PATRRs
We searched for PATRRs using the R package Biostring (v2.70.2) and its function "matchPattern" by detecting all positions of "AT" dinucleotides in each haplotype. Then we tiled each haplotype into 100 bp long bins and counted how many AT dinucleotides were in each bin. For each bin, we transformed AT dinucleotide counts into z-score and kept only those bins with z-score ≥ 3.29 (CI: 99.9%). After this, we collapsed neighboring bins into continuous ranges using the R package GenomicRanges (v1.54.1) and its function "reduce". We further removed all singleton ranges of 100 bp in size. Lastly, we collapsed all neighboring ranges within 10 kbp distance into the final set of PATRRs ranges. In the next step, we used these PATRR locations to divide the duplicon structure of each haplotype at the PATRR positions. We summarized these into five nonredundant building blocks (Supplementary Fig. 9) and searched for the locations of these five building blocks in each haplotype. For this, we gave each duplicon a unique number that was either positive or negative for direct and reverse orientation, respectively. We then recorded the best match of each of the five duplicon structures in each haplotype. This was done by measuring the distance between the numeric codes of each building block at all possible positions in each haplotype.

## Clustering LCRA by SD repeat annotation
We started by extracting FASTA sequences from the defined LCRA boundaries as described in "Definition of continuous LCR blocks." As mentioned above, we map all five SD repeat units to each LCRA haplotype ($n = 133$). Then we assigned each SD repeat unit a unique numeric identifier. This identifier is either a positive or a negative number in order to distinguish forward and reverse-mapped repeat units. We compared SD repeat annotation for all possible pairs of 133 haplotypes as pairs of numeric codes. The distance between each pair of numeric codes was calculated using the "seq_dist" function from the stringdist (v0.9.12) R package with the parameter "method" set to "osa". All pairwise distances are organized into the distance matrix, which is then used to construct the UPGMA tree using the function "upgma" from the R package phangorn (v2.11.1)[63]. We arrived at 63 unique haplogroups reported by the "cuttree" function (set parameter: $h = 0.5$) of the base R package stats.

## Evolutionary history of SD repeat units
The evolution of repeat segments was analyzed with additional NHPs, including the six great apes genomes[23] and a macaque genome[24]. The phylogeny of repeat segments was analyzed in two ways: first with the consensus sequence of each repeat unit (25, 28, 105, and 120 kbp) conserved up to chimpanzee, and secondly, with smaller individual duplicons defined by syntenic alignment blocks compared to macaque (T2T-MFA8v1.0)[24]. The alignment to outgroup macaque and NHP genomes was performed by minimap (v2.26)[60] using the following parameter to locate orthologous copies: `-cx asm20 --secondary=yes --eqx`. MSAs of the orthologous copies of repeat segments were performed using MAFFT (v7.525)[64] with the "--auto" option. For the phylogeny of individual duplicon units, the sequence corresponding to each unit was retained by trimming with Gblocks (v0.91b). A maximum likelihood tree was reconstructed using IQtree (v2.3.6)[65] with 1000 bootstrap. The tree was also outgrouped with single-copy orthologs from the macaque genome, and divergence time was estimated by recalibration based on divergence to the siamang and macaque genomes (19.5 and 28.8 MYA, respectively) or chimpanzee genome (6.4 MYA), following the TimeTree database[66].

## Mapping gene annotation on top of each haplotype
Gene annotations used in this study are reported with respect to the T2T-CHM13 reference and were obtained from a file 'chm13v2.0_RefSeq_Liftoff_v5.1.gff3.gz' at: https://s3-us-west-2.amazonaws.com/human-pangenomics/index.html?prefix=T2T/CHM13/assemblies/annotation/. We read the annotation file using the readGFF function from the R package rtracklayer (v1.62.0). Next, we extracted the genomic region for all gene boundaries as well as all exons for each gene. We extracted the FASTA sequence from all these regions and mapped them onto each of the haplotypes ($n = 135$). We used minimap2 (v2.24)[60] to map whole gene sequences (including introns) with the following parameters: `-x asm20 -c --eqx --secondary=no`. FASTA sequence from gene exons was mapped with minimap2 (v2.24)[60] using the following parameters: `-x splice -c --eqx --secondary=yes`.

## Analysis of public Iso-Seq datasets
We re-analyzed a collection of 1.4 billion full-length cDNA reads (Iso-Seq) from Dishuck et al.[67]. Briefly, Iso-Seq reads were extracted based on alignment to 22q11.2 or paralogous regions of T2T-CHM13 reference and re-aligned to each assembled haplotype with minimap2 (v2.26)[60]. We transferred GENCODE v44 gene models to each haplotype with Liftoff (v1.6.3)[68]. To generate de novo gene models, only reads with ≥99.9% identity to the best matching haplotype that uniquely map to a single paralog (at least 2 bp alignment delta) were used as input to PacBio Pigeon and SQANTI3 (v5.2)[69]. Tissue expression analyses considered only reads aligning with ≥99.9% identity to their best-matching haplotype (Supplementary Note 2).

## Mapping inversion breakpoints
To map inversion breakpoints between LCRA and LCRD in sample HG01888-H1, we first selected a human sample carrying a direct haplotype with a similar DupMasker profile (HG00621-H2). We next aligned the direct haplotype (HG00621-H2) to the inverted haplotype (HG01888-H1) using minimap2 (v2.24)[60] using the following parameters: `-x asm20 -c --eqx --secondary=no`. We projected positions of 25 kbp repeat units and PATRRs to both direct and inverted haplotypes. We then defined the putative location of the inversion breakpoint as a region encompassing both 25 kbp repeat unit and adjacent PATRR in proximal (LCRA) and distal (LCRD) regions of direct and inverted haplotypes. FASTA sequence from these defined regions was extracted, orientation synchronized, and an MSA constructed using the R package DECIPHER (v3.2.0, function "AlignSeqs")[62]. We extracted paralog-specific variants (PSVs) and defined the inversion

breakpoint as a region where inverted haplotypes (HG01888-H1) partly match proximal and distal PSVs of the direct haplotype (HG00621-H2) (Supplementary Fig. 39). For more details see Porubsky et al.[34].

## Inversion genotyping within 1KGP cohort

For this analysis, we first identified single-nucleotide variants (SNVs) in the assembled inverted (A-D inversion, haplotype 1) and direct haplotype for the 22q11.2 region in sample HG01888. PAV (v2.3.4)[20] was used to call variants with respect to both the GRCh38 and T2T-CHM13 reference genomes. The genotyping was done against a diversity panel of 3202 1KGP samples available for both reference genomes[37,38]. For genotyping we selected SNV positions present in both PAV as well as the 1KGP callset. Then we compared HG01888 haplotypes to all 3202 samples (6404 haplotypes) and calculated the fraction of matching SNVs between PAV and the 1KGP allele. We selected samples whose fraction of matching SNVs was ≥5 standard deviations ($z$-score ≥ 5) from the population mean (average fraction of matching SNVs across all comparisons) as candidate samples to carry the inversion haplotype identified in HG01888. This approach was repeated for other A-D inversion carriers (NA19315, HG03239, HG01947, and HG01489) (Supplementary Fig. 37) as well as for other inversion breakpoints (A-B, B-D, and C-D) in HG03471, HG01695, HG01175, and HG03139 (Supplementary Fig. 38).

## Detection of inverted duplications within LCRA and D

In order to detect other commonly inverted regions outside of large A-D, A-B, and B-D inversions, we searched for inverted alignments in available human haplotypes with respect to a single reference 22q11.2 haplotype. We chose as a reference haplotype HG01890-H1 of African ancestry because this haplotype carries exactly one copy of the 155 kbp repeat unit (105 kbp plus two 25 kbp units) in both LCRA and D. We used minimap2 (v2.24)[60] to align all uninverted human haplotypes ($n = 133$) with respect to HG01890-H1 with the following parameters: `-x asm20 -c --eqx --secondary=no`. We then processed alignment of each human haplotype as follows: We filter out query-to-target alignments that map between non-syntenic LCR blocks. Next, we retained all inverted alignments ≥10 kbp in length. We collapsed gaps between alignments ≤1 kbp into a single region. Then we grouped alignments based on the required reciprocal overlap of ≥60% (Supplementary Fig. 36), and for each alignment group, we report inversion boundaries as a range that is covered by half and more alignments in a given group. Lastly, we only retain those inversion ranges supported by at least five and more inverted alignments (Supplementary Data 4).

## Analysis of intra- and interchromosomal NAHRs at 22q11.2

In this analysis we first defined the positions of all 105 kbp repeat units by aligning the HOR annotation to each haplotype as described in 'Definition of shared SD repeat units between LCRA and D'. We start by tracking the sequence identity and orientation of all intrachromosomal (cis) interactions between LCRA and D within human haplotypes ($n = 133$ plus the T2T-CHM13 reference). For each haplotype we define all possible pairs of the 105 kbp repeat unit between LCRA and D and report their relative orientation (forward - direct or reverse - inverted) and sequence identity. We mark haplotypes where ≥75% of all LCRA and D interactions (SD pairs) are in reverse or forward orientation as either protected or predisposed to 22q11.2DS, respectively.

To boost our predictions regarding the deletion predisposition at 22q11.2, we expanded the analysis to include samples from both the HGSVC ($n = 65$) and HPRC2 ($n = 232$) in order to find as many samples as possible with both parental haplotypes fully and accurately assembled. In total, among the HGSVC and HPRC2 assemblies, we identified 233 (41 from HGSVC and 192 from HPRC2) samples where both parental haplotypes are assembled in a single continuous contig. In this analysis, we focus on all possible interchromosomal (trans) interactions between all copies of the 105 kbp repeat unit present in LCRA and

D that could be implicated in 22q11.2 deletion formation via NAHR between parental homologs. Again, we mark haplotypes where ≥75% of all LCRA and D interactions (SD pairs) are in reverse or forward orientation as either protected or predisposed to 22q11.2DS, respectively.

## Mapping A-D deletion breakpoints with assemblies

We mapped A-D deletion breakpoints in three family duos (AD009, AD010, and AD013) and one family trio, such that we first created all-versus-all alignments between the proband haplotype that carries the A-D deletion and both parental haplotypes (transmitting parent). We used minimap2 (v2.24) with the following parameters: `-x asm20 -c --eqx --secondary=no`. We located redundant (overlapping) alignments between proband and parental haplotypes at the deletion breakpoint, followed by extracting the FASTA sequence from these regions across all haplotypes (proband and parental haplotypes) (Supplementary Figs. 43 and 44). We synchronized the orientation of these sequences and constructed an MSA using the R package DECI-PHER (v3.2.0, function "AlignSeqs")[62]. We then extracted PSVs and defined the deletion breakpoint as a region where the deleted haplotype partly matches PSVs from either parental haplotype. For more details on mapping changepoints in PSV profiles see Porubsky et al.[34].

## Evaluation of alignment identity among 105 kbp repeats

We determined that putative A-D deletion breakpoints map within the 105 kbp SD repeat unit. Therefore, we set out to define the region of the highest identity within all possible SD pairs between LCRA and D. To achieve this, we aligned each SD pair to each other using minimap2 (v2.28)[60] with the following parameters: `-x asm20 -c --eqx --secondary=no`. We followed this by binning each pairwise alignment of the 105 kbp segment between LCRA and D into 1 kbp-long bins. For each bin, we calculated the percentage of matched base pairs. Finally, we selected a continuous set of bins with the percentage of matched base pairs ≥99.5% and collapsed into a single region of high sequence identity. We also tracked the relative orientation of these high-identity regions between LCRA and D as either forward or reverse oriented. With this, we were able to report that the highest identity (≥99.5%) regions reside in two locations (around 50 kbp and 74 kbp) within 105 kbp repeat unit (Fig. 6e).

## Reporting summary

Further information on research design is available in the Nature Portfolio Reporting Summary linked to this article.

## Data availability

HGSVC assemblies ($n = 65$) used in this study were reported by Logsdon et al.[22] under NCBI BioProject ID: PRJEB83624. HPRC1 (release 1) assemblies ($n = 47$) used in this study were reported by Liao et al.[21] under NCBI BioProject ID: PRJNA730822. Additional HPRC2 (release 2) assemblies ($n = 232$) are available via NCBI under BioProject ID: PRJNA730823 or via GitHub [https://github.com/human-pangenomics/hprc_intermediate_assembly]. LRS data for 1KGP samples HG01888 and NA19315 together with Strand-seq data ($n = 13$) generated for genotyping purposes of HG01888 inversion are available via European Nucleotide Archive (ENA) under BioProject ID: PRJEB91688. An assembly for sample HG03471 was released by Hallast et al.[70] [https://ftp.1000genomes.ebi.ac.uk/vol1/ftp/data_collections/HGSVC3/working/20230412_sigY_assembly_data/]. LRS data for clinical parent–child duos (AD009-father, AD009-proband, AD010-father, AD010-proband, AD013-mother, and AD013-proband) and for the Coriell trio family (NA10382, NA10393, and NA10384) are available via European Genome-phenome Archive (EGA) under accession number: EGAD50000002367. To ensure the privacy of clinical patient data, access is restricted and requires a formal application via the EGA website. Researchers can only gain access to the data with the

permission of the UZ/KU Leuven Data Access Committee (DAC). The data transfer will be covered by a Data Access Agreement (DAA), which defines the terms and conditions of use. The DAA is created and provided by the DAC and must be signed by the individual, wishing to access the given dataset. FASTA sequences for 135 human and 12 NHP 22q11.2 haplotypes are available via Zenodo (doi: 10.5281/zenodo.17950312 [https://zenodo.org/records/17950312]). In addition, FASTA sequences for 233 samples (466 haplotypes) used for the NAHR analysis along with FASTA sequences of all five SD repeat units defined in this manuscript are available via Zenodo (doi: 10.5281/zenodo.17950312 [https://zenodo.org/records/17950312]). Processed Strand-seq pool data for 279 1KGP samples used for inversion genotyping at 22q11.2 are available as both UCSC Genome Browser formatted BED files and BAM files (via Zenodo; https://doi.org/10.5281/zenodo.17950312 [https://zenodo.org/records/17950312]). Source data are provided with this paper.

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

## Acknowledgements

We thank Tonia Brown for editing and preparation of this manuscript. We also thank Mianne Lee and Trang Nguyen for technical assistance with preparation and sequencing of Strand-seq libraries. This work was supported, in part, by US National Institutes of Health (NIH) grants HG010169, HG010971, and HG007497 to E.E.E. Strand-seq work in the Landsdorp lab is supported by grants from the Terry Fox Research Institute (Program Project Grant #1074), the Canadian Institutes of Health Research (CIHR grants PJT-159787 and PWUZ GR028457), the Canada Foundation for Innovation (CFI grants #40044 and #43153), Genome Canada and Genome BC (project 323ECC), and the BC Cancer Foundation. Additional funding support is acknowledged from the Fonds Wetenschappelijk Onderzoek (grant G0A2622N) to J.R.V., KU Leuven (C14/22/125) to J.R.V., and Stichting Marguerite-Marie Delacroix (RVC/B-534) to S.M. Last, we would like to acknowledge the National Genome Research Institute (NHGRI) for funding the following grants supporting the creation of the human pangenome reference: U41HG010972, U01HG010971, U01HG013760, U01HG013755, U01HG013748, U01HG013744, R01HG011274, and the Human Pangenome Reference Consortium (BioProject ID: PRJNA730823). The content is solely the responsibility of the authors and does not necessarily represent the official views of the NIH. E.E.E. is an investigator of the Howard Hughes Medical Institute. The following cell lines were obtained from the NIGMS Human Genetic Cell Repository at the Coriell Institute for Medical Research: GM01888, GM19315, GM10382, GM10383, GM10384, GM01883, GM02562, GM02563, GM02611, GM03112, GM18870, GM18872, GM19152, GM19154, GM19221, GM19223, GM01102, and GM01103. This article is subject to HHMI's Open Access to Publications policy. HHMI lab heads have previously granted a nonexclusive CC BY 4.0 license to the public and a sublicensable license to HHMI in their research articles. Pursuant to those licenses, the author-accepted manuscript of this article can be made freely available under a CC BY 4.0 license immediately upon publication.

## Author contributions

D.P., J.R.V., and E.E.E. conceptualized the study. V.T., D.D.C., T.Y.L., P.H., J.O.K, and P.M.L. Strand-seq data generation. D.P. Strand-seq data analysis. K.M.M., K.H., D.P., N.K., HPRC, and E.E.E. Long-read data and assembly generation. N.K., T.Y.L., and W.T.H. Data wrangling and analysis support. E.S., M.S.S., S.M., A.S., J.B., and J.R.V. Patient data collection. D.P., D.Y., E.S., P.C.D., N.D., and E.E.E. Data analysis and interpretation. D.P., D.Y., E.S., and E.E.E. developed main figures and supplementary information. D.P., D.Y., P.C.D., and E.E.E. manuscript writing. E.E.E. Supervised the study.

## Competing interests

E.E.E. is a scientific advisory board (SAB) member of Variant Bio, Inc. D.P. and J.O.K. have previously disclosed a patent application (no.

EP19169090) relevant to Strand-seq. P.M.L. is a founding shareholder of Repeat Diagnostic, Inc. and in Evident Genomics, Inc. He is listed as an inventor in US patent US-20250146052-A1. All other authors declare no competing interests.

## Additional information

## Human Pangenome Reference Consortium

Derek Albracht[9], Ivan A. Alexandrov[10], Jamie Allen[11], Alawi A. Alsheikh-Ali[12], Nicolas Altemose[13], Casey Andrews[14], Dmitry Antipov[15], Lucinda Antonacci-Fulton[9], Mobin Asri[16], Marcelo Ayllon[1], Jennifer R. Balacco[17], Floris P. Barthel[18], Edward A. Belter Jr[9], Halle D. Bender[16], Andrew P. Blair[16], Davide Bolognini[19], Katherine E. Bonini[20], Christina Boucher[21], Guillaume Bourque[22,23,24], Silvia Buonaiuto[25], Shuo Cao[25], Andrew Carroll[26], Ann M. Mc Cartney[16], Monika Cechova[16], Mark J. P. Chaisson[27], Pi-Chuan Chang[26], Xian Chang[16], Jitender Cheema[11], Haoyu Cheng[28], Claudio Ciofi[29], Hiram Clawson[16], Sarah Cody[9], Vincenza Colonna[25], Holland C. Conwell[30], Robert Cook-Deegan[31], Mark Diekhans[16], Maria Angela Diroma[29], Daniel Doerr[32,33,34], Zheng Dong[14], Danilo Dubocanin[13], Richard Durbin[35,36], Jana Ebler[32,37], Evan E. Eichler[1,8,38], Jordan M. Eizenga[16], Parsa Eskandar[16], Eddie Ferro[21], Anna-Sophie Fiston-Lavier[39,40], Sarah M. Ford[30], Willard W. Ford[41], Giulio Formenti[17], Adam Frankish[11], Mallory A. Freeberg[11], Qichen Fu[14], Stephanie M. Fullerton[42], Robert S. Fulton[9], Yan Gao[43], Gage H. Garcia[1], Obed A. Garcia[44], Joshua M. V. Gardner[16], Shilpa Garg[45], Erik Garrison[25], Nanibaa' A. Garrison[46,47,48], John E. Garza[9], Margarita Geleta[49], Mohammadmersad Ghorbani[50], Tina A. Graves-Lindsay[9], Richard E. Green[30], Cristian Groza[51], Bida Gu[27], Andrea Guarracino[18,25], Melissa Gymrek[52], Maximilian Haeussler[16], Leanne Haggerty[11], Ira M. Hall[53,54], Nancy F. Hansen[15], Yue Hao[18], Mohammad Amiruddin Hashmi[12], David Haussler[16], Prajna Hebbar[16], Peter Heringer[32,33,34], Glenn Hickey[16], Todd L. Hillaker[16], S. Nakib Hossain[11], Neng Huang[43,55], Sarah E. Hunt[11], Toby Hunt[11], Alexander G. Ioannidis[13,16], Nafiseh Jafarzadeh[16], Nivesh Jain[17], Erich D. Jarvis[17,38], Maryam Jehangir[18], Juan Jiang[14], Eimear E. Kenny[20], Juhyun Kim[15], Bonhwang Koo[17], Sergey Koren[15], Milinn Kremitzki[9,14], Charles H. Langley[56], Ben Langmead[57], Heather A. Lawson[14], Daofeng Li[14], Heng Li[43,55], Wen-Wei Liao[53,54], Jiadong Lin[1], Tianjie Liu[14], Glennis A. Logsdon[58], Ryan Lorig-Roach[16], Jonathan LoTempio Jr[59], Hailey Loucks[16], Jane E. Loveland[11], Jianguo Lu[60], Shuangjia Lu[53,54], Julian K. Lucas[16], Walfred Ma[27], Juan F. Macias-Velasco[9,14,61], Kateryna D. Makova[62], Maximillian G. Marin[43,55], Christopher Markovic[9], Tobias Marschall[32,37], Franco L. Marsico[25], Fergal J. Martin[11], Mira Mastoras[16], Capucine Mayoud[39], Brandy McNulty[16], Jack A. Medico[17], Julian M. Menendez[16], Karen H. Miga[16], Anna Minkina[63], Matthew W. Mitchell[64], Saswat K. Mohanty[65], Younes Mokrab[50,66,67], Jean Monlong[68], Shabir Moosa[50], Avelina Moreno-Ochando[69,70], Shinichi Morishita[71], Jonathan M. Mudge[11], Katherine M. Munson ®[1], Njagi Mwaniki[72], Nasna Nassir[12], Chiara Natali[29], Shloka Negi[16], Lingbin Ni[1], Adam M. Novak[16], Pilar N. Ossorio[73], Chie Owa[71], Sadye Paez[17], Benedict Paten[16], Clelia Peano[19,74], Adam M. Phillippy[15], Brandon D. Pickett[15], Laura Pignata[25], Nadia Pisanti[72], David Porubsky ®[1,2], Pjotr Prins[25], Anandi Radhakrishnan[16], T. Rhyker Ranallo-Benavidez[18], Brian J. Raney[16], Mikko Rautiainen[75], Alessandro Raveane[19], Luyao Ren[1,38], Arang Rhie[15], Fedor Ryabov[76,77], Samuel Sacco[30], Farnaz Salehi[25], Michael C. Schatz[57,78], Laura B. Scheinfeldt[79], Aarushi Sehgal[41], William E. Seligmann[30], Mahsa Shabani[80], Kishwar Shafin[26], Shadi Shahatit[39], Ruhollah Shemirani[20], Vikram S. Shivakumar[57], Swati Sinha[11], Jouni Sirén[16], Linnéa Smeds[65], Steven J. Solar[15], Marco Sollitto[17,29], Nicole Soranzo[19,35,81], Andrew B. Stergachis[1,63], Marie-Marthe Suner[11], Yoshihiko Suzuki[71], Arda Söylev[32,37], Ahmad Abou Tayoun[82,83], Jack A. S. Tierney[11], Chad Tomlinson[9], Francesca Floriana Tricomi[11], Mohammed Uddin[12,84], Matteo Tommaso Ungaro[30,85], Rahul Varki[21], Flavia Villani[25], Ivo Violich[16], Mitchell R. Vollger[63], Brian P. Walenz[15], Charles Wang[86], Lisa E. Wang[20], Ting Wang[9,14,54], Aaron M. Wenger[87],

Conor V. Whelan[17], Zilan Xin[14], Zheng Xu[14], Kai Ye[88], DongAhn Yoo [1], Wenjin Zhang[14], Ying Zhou[43], Xiaoyu Zhuo[14] & Giulia Zunino[19]

[9]McDonnell Genome Institute, Washington University School of Medicine, St. Louis, MO, USA. [10]Department of Human Molecular Genetics and Biochemistry, Faculty of Medical and Health Sciences, Tel Aviv University, Tel Aviv, Israel. [11]European Molecular Biology Laboratory, European Bioinformatics Institute (EMBL-EBI), Wellcome Genome Campus, Hinxton, Cambridge, UK. [12]Center for Applied and Translational Genomics (CATG), Mohammed Bin Rashid University of Medicine and Health Sciences, Dubai Health, Dubai, UAE. [13]Department of Genetics, Stanford University, Palo Alto, CA, USA. [14]Department of Genetics, Washington University School of Medicine, St. Louis, MO, USA. [15]Genome Informatics Section, Center for Genomics and Data Science Research, National Human Genome Research Institute, National Institutes of Health, Bethesda, MD, USA. [16]UC Santa Cruz Genomics Institute, University of California, Santa Cruz, CA, USA. [17]The Vertebrate Genome Laboratory, The Rockefeller University, New York, NY, USA. [18]Bioinnovation and Genome Sciences, The Translational Genomics Research Institute (TGen), Phoenix, AZ, USA. [19]Human Technopole, Milan, Italy. [20]Institute for Genomic Health, Icahn School of Medicine at Mount Sinai, New York, NY, USA. [21]Department of Computer and Information Science and Engineering, University of Florida, Gainesville, FL, USA. [22]Canadian Center for Computational Genomics, McGill University, Montréal, QC, Canada. [23]Department of Human Genetics, McGill University, Montréal, QC, Canada. [24]Victor Phillip Dahdaleh Institute of Genomic Medicine, Montréal, QC, Canada. [25]Department of Genetics, Genomics and Informatics, University of Tennessee Health Science Center, Memphis, TN, USA. [26]Google LLC, Mountain View, CA, USA. [27]Quantitative and Computational Biology, University of Southern California, Los Angeles, CA, USA. [28]Department of Biomedical Informatics and Data Science, Yale School of Medicine, New Haven, CT, USA. [29]Department of Biology, University of Florence, Sesto Fiorentino, FI, Italy. [30]Department of Ecology and Evolutionary Biology, University of California, Santa Cruz, CA, USA. [31]Arizona State University, Consortium for Science, Policy & Outcomes, Washington, DC, USA. [32]Center for Digital Medicine, Heinrich Heine University Düsseldorf, Düsseldorf, NRW, Germany. [33]Department for Endocrinology and Diabetology at the Medical Faculty and University Hospital Düsseldorf, Heinrich Heine University Düsseldorf, Düsseldorf, NRW, Germany. [34]Paul-Langerhans-Group Computational Diabetology, German Diabetes Center (DDZ) and Leibniz Institute for Diabetes Research, Düsseldorf, NRW, Germany. [35]Wellcome Sanger Institute, Genome Campus, Hinxton, UK. [36]Department of Genetics, University of Cambridge, Cambridge, UK. [37]Institute for Medical Biometry and Bioinformatics, Medical Faculty and University Hospital Düsseldorf, Heinrich Heine University, Düsseldorf, NRW, Germany. [38]Howard Hughes Medical Institute, Chevy Chase, MD, USA. [39]ISEM, Univ Montpellier, CNRS, IRD, Montpellier, France. [40]Institut Universitaire de France, Paris, France. [41]Department of Computer Science and Engineering, University of California San Diego, La Jolla, CA, USA. [42]Department of Bioethics & Humanities, University of Washington School of Medicine, Seattle, WA, USA. [43]Department of Data Science, Dana-Farber Cancer Institute, Boston, MA, USA. [44]Department of Anthropology, University of Kansas, Lawrence, KS, USA. [45]School of Health Sciences, University of Manchester, Manchester, UK. [46]Traditional, ancestral and unceded territory of the Gabrielino/Tongva peoples, Institute for Society & Genetics, University of California, Los Angeles, Los Angeles, CA, USA. [47]Traditional, ancestral and unceded territory of the Gabrielino/Tongva peoples, Institute for Precision Health, David Geffen School of Medicine, University of California, Los Angeles, Los Angeles, CA, USA. [48]Traditional, ancestral and unceded territory of the Gabrielino/Tongva peoples, Division of General Internal Medicine & Health Services Research, David Geffen School of Medicine, University of California, Los Angeles, Los Angeles, CA, USA. [49]Department of Electrical Engineering and Computer Science, University of California, Berkeley, Berkeley, CA, USA. [50]Medical and Population Genomics Lab, Sidra Medicine, Doha, Qatar. [51]Montreal Heart Institute, Montréal, QC, Canada. [52]Department of Pediatrics, University of California San Diego, La Jolla, CA, USA. [53]Center for Genomic Health, Yale University School of Medicine, New Haven, CT, USA. [54]Department of Genetics, Yale University School of Medicine, New Haven, CT, USA. [55]Department of Biomedical Informatics, Harvard Medical School, Boston, MA, USA. [56]Department of Evolution and Ecology and the Center for Population Biology, University of California, One Shields, Davis, CA, USA. [57]Department of Computer Science, Johns Hopkins University, Baltimore, MD, USA. [58]Department of Genetics, Epigenetics Institute, Perelman School of Medicine, University of Pennsylvania, Philadelphia, PA, USA. [59]Department of Pediatrics, Division of Genetics, School of Medicine, University of California, Irvine, CA, USA. [60]Sun Yat-sen University, Guangzhou, China. [61]Edison Family Center for Genome Sciences & Systems Biology, Washington University School of Medicine, St. Louis, MO, USA. [62]Department of Biology and Center for Medical Genomics, Penn State University, University Park, PA, USA. [63]Division of Medical Genetics, Department of Medicine, University of Washington School of Medicine, Seattle, WA, USA. [64]The Jackson Laboratory for Genomic Medicine, Farmington, CT, USA. [65]Department of Biology, Penn State University, University Park, PA, USA. [66]Department of Biomedical Science, College of Health Sciences, Qatar University, Doha, Qatar. [67]Department of Genetic Medicine, Weill Cornell Medicine-Qatar, Doha, Qatar. [68]IRSD - Digestive Health Research Institute, University of Toulouse, INSERM, INRAE, ENVT, UPS, Toulouse, France. [69]MATCH biosystems, S.L., Elche, Spain. [70]Universidad Miguel Hernández de Elche, Elche, Spain. [71]Department of Computational Biology and Medical Sciences, The University of Tokyo, Kashiwa, Chiba, Japan. [72]Department of Computer Science, University of Pisa, Pisa, Italy. [73]Law School, University of Wisconsin-Madison, Madison, WI, USA. [74]Institute of Genetics and Biomedical Research, UoS of Milan, National Research Council, Milan, Italy. [75]Institute for Molecular Medicine Finland, Helsinki Institute of Life Science, University of Helsinki, Helsinki, Finland. [76]The Center for Bio- and Medical Technologies, Moscow, Russia. [77]Centre for Biomedical Research and Technology, HSE University, Moscow, Russia. [78]Department of Biology, Johns Hopkins University, Baltimore, MD, USA. [79]Coriell Institute for Medical Research, Camden, NJ, USA. [80]University of Amsterdam, Amsterdam, The Netherlands. [81]School of Clinical Medicine, University of Cambridge, Cambridge, UK. [82]Center for Genomic Discovery, Mohammed Bin Rashid University, Dubai Health, Dubai, UAE. [83]Dubai Health Genomic Medicine Center, Dubai Health, Dubai, UAE. [84]GenomeArc Inc, Mississauga, ON, Canada. [85]Department of Biology and Biotechnologies "Charles Darwin", University of Rome "La Sapienza", Rome, Italy. [86]Center for Genomics, Loma Linda University School of Medicine, Loma Linda, CA, USA. [87]PacBio, Menlo Park, CA, USA. [88]The first affiliated hospital of Xi'an Jiaotong University, Xi'an Jiaotong University, Xi'an, Shaanxi, China.

