## [Transparent Peer Review file · Nature Communications]

Population differences of chromosome 22q11.2 duplication structure predispose differentially to microdeletion and inversion

Corresponding Author: Professor Evan Eichler

Version 0:

Reviewer comments:

Reviewer #1

(Remarks to the Author)

This is a very well-written manuscript, which presents a completely sequence-based, nucleotide-level analysis and map of the highly complex and identical segmental duplications (SDs) in chromosome 22q11.2. They have used complete chromosome 22q11.2 sequence, generated as part of the T2T consortium, to analyze 135 diverse human haplotypes and 12 non-human primate haplotypes. In addition, they have analyzed patient families with the T2T sequencing technologies to identify rearrangement hotspots.

Overall, it was a pleasure to read this manuscript as it clearly presents very complex data, making it very easy for the reader to follow the authors' results and interpretation.

This is the first completely sequence-based approach, which allows them to present the structure and make-up of duplicated modules within the LCRs with base-pair level resolution. Their findings generally confirm the results and predictions made by several previous studies using techniques like optical mapping and fiber-FISH.

Thus, the authors have exploited their access to the T2T sequence data to confirm previous knowledge and findings about the 22q11.2 LCRs and their complexity. The sequence data from their analyses will be useful to the field, provided that it is made available in a useful form and a timely manner to the larger scientific community.

The following are issues and questions for the authors to address in support of their conclusions.

Major Issues:

1) The authors present the analysis of 135 diverse haplotypes spanning the 22q11.2 region across all LCRs in a single contig. However, it isn't clear from the Results section how many of the presented haplotypes included those that were both haplotypes from the same individual.

2) Presumably, there are 440 potential haplotypes from 220 phased genome assemblies. But the fact that they only have 135 complete haplotypes raises the question about the accuracy of the assemblies, especially if both haplotypes in a given individual were not assembled into complete contigs.

This is a very critical issue in the accurate assembly of SD/LCR containing regions. Unless both haplotypes from a given individual are assembled into single contigs without any unassigned sub-contigs, it is difficult to assess the accuracy of the assembly.

3) This raises further concerns about the results and conclusions presented here which were derived from the assembled haplotypes.

Notably, NAHR is likely to be affected by the SD/LCR structure on both haplotypes in the Parent of Origin. Thus, any conclusions about African ancestry genomes (versus other ancestries) being more predisposed to one type of

rearrangement versus another based on single haplotypes from any individual seems premature.

Thus, the entire Result section titled "Predisposition to 22q11.2DS..." on Page 12 as well as any resulting conclusions in the Discussion need to be reconsidered and restated based on this limitation of their assemblies and data used.

4) Similarly, it is difficult to assess the accuracy of the assemblies in the patient family trios as these data are currently not available. It would be useful for the reader to understand how the authors were able to assemble these haplotypes with confidence when the T2T consortium genomes appear to be partially (or entirely) incompletely assembled across the 22q11 region.

5) The authors may have inadvertently forgotten to acknowledge that the localization of deletion breakpoints (Page 14) within what they call the 105 Kb repeat unit was originally shown by Demaerel et al (Ref 25), who refer to that region as SD22-4. Granted, that the current work provides sequence level resolution to confirm the findings of Demaerel et al. and others.

6) The authors seem to downplay the previous work of others when it is clear that many of their sequence resolved results were already predicted or known based on previous studies. This includes a large body of work on PATRRs in 22q11 rearrangements, and breakpoints localized in the SDs/LCRs on 22q11. Proper acknowledgement of these studies would be beneficial and help the reader understand the progression of the work on this difficult region of the genome and the authors' contribution to the overall deciphering of the complexity of 22q11.2.

Minor Issue:

1) There is some gibberish on line 5, page 7.

Reviewer #2

(Remarks to the Author)

The manuscript by Coughlin et al. presents a comprehensive study of the structural haplotype architecture in the 22q11.2 region, combining PacBio HiFi long-read data from diverse global populations. The authors link specific structural haplotypes with predisposition to recurrent rearrangements, specifically, LCR22A–D deletions and inversions, and find striking population differences. This is an elegant and ambitious study with substantial implications for both genome biology and clinical genetics. However, several aspects of figure presentation, data summarization, and quality control reporting could be improved to enhance clarity and reproducibility.

Major Points

1. Figure design and visual clarity

Figures contain a large number of data points, tracks, or labels crammed into a small space. While they are information-rich and visually appealing, many of them are too small, overly dense, or lack clear labeling, which significantly reduces their accessibility. The font size of labels is consistently small.

Suggestion: Increase font size and streamline visual elements. Consider separating complex panels into multiple figures or using simplified summary schematics alongside detailed plots.

A summary table or flowchart explaining the key structural haplotypes (such as SD22-3 and SD22-4 configurations) and their predisposition to NAHR-mediated events across populations would help guide the reader through the results.

2. Missing details on quality control and data filtering

The resolution and phasing presented in the study are impressive, but the QC thresholds and filtering steps used to achieve this level of accuracy are not clearly reported.

For instance, what coverage was required? What were the minimum read lengths? How many of the initial samples failed assembly or phasing, and why?

These details are important for readers hoping to replicate or build on this work.

Suggestion: Add a QC and data summary table, reporting per-sample metrics (read N50, coverage, contig length across the 22q11.2 region, etc.), success rates of phasing/assembly, and failure reasons.

3. Summarization of results could be improved

The main text sometimes moves quickly between specific findings without offering overarching summaries. For example, the description of inversion susceptibility and differences between populations (e.g., between African and European haplotypes) is fascinating but could be better synthesized.

A summary figure or model illustrating which structural features (e.g., SD orientation, duplication content) predispose to deletions vs inversions would make the core results more accessible to a broad readership.

4. Functional or clinical implications not fully developed

While the structural and evolutionary findings are strong, the potential clinical or mechanistic relevance could be emphasized further.

For example, how might these findings inform carrier screening, CNV risk stratification, or population-specific variant interpretation?

Would these structures affect mapping or CNV-calling in short-read data?

Minor Points

5. Use of color and labels in figures

Several figures use similar colors to distinguish critical structural features, making it hard to distinguish them in print or grayscale.

Suggestion: Use more distinct color palettes and add direct in-figure labeling or legends (rather than relying on external figure captions).

Reviewer #3

(Remarks to the Author)

Porubsky et al. describe population diversity and evolution and of four complex and unstable segmental duplication clusters (LCRA-D) in the 5 Mb Chromosome 22q11.2 deletion syndrome region. Using computational analyzes, including de novo genome assembly of long read PacBio HiFi and ONT whole genome sequencing and low-coverage Strand-seq data, they sequence-resolved 135 complete human 22q11.2 haplotypes from diverse ethnicities (African, American, East Asian, Southeast Asian, and European, 1000 Genome Project) and syntenic regions from six nonhuman primate species. Consistent with previous studies, LCRA was found to be most variable. They defined a 105 kb higher-order SD cassette flanked by 25 kb inverted repeats adjacent to PATRR sequences, as a major genomic instability driver that had emerged in the human-chimpanzee ancestral lineage and has expanded in humans less than 1.0 million years ago. They found that LCRA-D deletion breakpoints map to the 105 kb repeat unit whereas large rare inversion and inverted duplication breakpoints map to the 25 kb repeats. Corroborating previous clinical observations, African LCR-A haplotypes were found to be protecting against recurrent microdeletions, but more predisposing to genomic inversions because of the preponderance of inversely oriented repeats.

The manuscript is well-written and adds to the literature.

My major criticism is that the Authors present little molecular validation of their computational findings.

Introduction

This deletion, also known as DiGeorge or velocardiofacial syndrome

Please rephrase - a deletion is not a syndrome.

Page 4, Results

We initially evaluated the completeness of 220 phased genome assemblies generated as part of the HPRC18 and HGSVC19. Of these, 135 haplotypes were unrelated and assembled in a single continuous contig, passing our assembly quality validations²²

It is unclear to what extent these 135 haplotypes were assembled in the previous works PMID: 37165242 and 40702183.

What do they mean by "evaluation of the completeness"?

Page 5, Results

We assigned SDs to LCR regions...

Do they distinguish SDs from LCRs? In the Abstract, they state "segmental duplications (SDs) known as low copy repeats (LCRs).

Page 7, Results, para 2, lines 5-6

Illegible statement.

Page 17, Discussion

We find that inversion breakpoints cluster near PATRRs associated with the 25 kbp repeat unit

What do they mean by near? Please be more precise.

References 11, 15, 16, 17, 19, 36, 56, 59, 61 are incomplete or incorrect.

The manuscript is not double-spaced.

Version 1:

Reviewer comments:

Reviewer #1

(Remarks to the Author)

The authors have satisfactorily addressed the questions and concerns raised in the previous review of the manuscript. The clarifications provided along with the additional data and text have greatly improved the quality of the manuscript.

Reviewer #2

(Remarks to the Author)

The authors have responded to my comments, I find the manuscript improved and ready for publication.

Reviewer #3

(Remarks to the Author)

I have no further comments

RESPONSE TO REVIEWER COMMENTS

Reviewer #1 (Remarks to the Author):

This is a very well-written manuscript, which presents a completely sequence-based, nucleotide-level analysis and map of the highly complex and identical segmental duplications (SDs) in chromosome 22q11.2. They have used complete chromosome 22q11.2 sequence, generated as part of the T2T consortium, to analyze 135 diverse human haplotypes and 12 non-human primate haplotypes. In addition, they have analyzed patient families with the T2T sequencing technologies to identify rearrangement hotspots.

Overall, it was a pleasure to read this manuscript as it clearly presents very complex data, making it very easy for the reader to follow the authors' results and interpretation.

This is the first completely sequence-based approach, which allows them to present the structure and make-up of duplicated modules within the LCRs with base-pair level resolution. Their findings generally confirm the results and predictions made by several previous studies using techniques like optical mapping and fiber-FISH.

Thus, the authors have exploited their access to the T2T sequence data to confirm previous knowledge and findings about the 22q11.2 LCRs and their complexity. The sequence data from their analyses will be useful to the field, provided that it is made available in a useful form and a timely manner to the larger scientific community.

The following are issues and questions for the authors to address in support of their conclusions.

We thank the reviewer for thoughtful comments and useful suggestions to improve this manuscript.

Major Issues:

1) The authors present the analysis of 135 diverse haplotypes spanning the 22q11.2 region across all LCRs in a single contig. However, it isn't clear from the Results section how many of the presented haplotypes included those that were both haplotypes from the same individual.

Thank you for this question. Indeed, this is an important piece of information that was not reported. Now we report the total number of samples (57 out of 78, 73%) for which both haplotypes were fully and accurately assembled. We also added this information to **Supplementary Fig. 3** (see below).

Revised text (in bold):

To establish a human population diversity panel of 22q11.2, we evaluated the completeness of 110 phased genome assemblies (220 haplotypes) generated as part of

the HPRC1 (release 1)(Liao et al. 2023) and HGSCV(Logsdon et al. 2025), including two samples with reported large-scale inversions at 22q11.2. Of these, 135 haplotypes were unrelated, assembled in a single continuous contig, and passed our assembly quality validations(Vollger et al. 2019) (Supplementary Note 1, Methods). Among those, there were 78 diverse individuals of which 57 have both maternal and paternal haplotypes fully assembled. The sample set included representation of all human ancestries (AFR - African, AMR - American, EAS - East Asian, SAS - Southeast Asian and EUR - European) with comparable numbers of haplotypes of African (n=59, 43.7%) and non-African ancestry (n=76, 57.3%) (Fig. 1c, Supplementary Fig. 3).

Supplementary Figure 3: Summary of complete assemblies of the 22q11.2 region.

a) Barplot showing the number of complete assemblies (assembled in a single contig) and NucFreq validated assemblies (Valid, TRUE - green; invalid, FALSE - orange) stratified by assembly source (HGSVC, HPRC1 - release 1). **b)** A stacked barplot showing the proportion of all evaluated assemblies being assembled in a single contig and validated by NucFreq. **c)** Barplot showing the counts for all NucFreq validated assemblies stratified by ancestry. **d)** A barplot showing the number of samples with both maternal and paternal haplotypes assembled (n=57) and those with only one haplotype assembled.

Note: Counts in a) and b) do not include four haplotypes from two samples (HG01888 and HG03471) that carry a large-scale inversion over the 22q11.2 loci; thus, the total count reported here is 131. Counts in c) and d) consider the full panel of 135 diverse haplotypes.

2) Presumably, there are 440 potential haplotypes from 220 phased genome assemblies. But the fact that they only have 135 complete haplotypes raises the question about the accuracy of the assemblies, especially if both haplotypes in a given individual were not assembled into complete contigs.

We subjected each of the genomes included in this analysis to rigorous assembly QC metrics, excluding those that failed QC, and added a detailed supplementary note (see detailed description in Supplementary Note 1 below). In short, we estimate that the assemblies we present are ~99.99% accurate. The reviewer, however, raises another potential point of confusion that needs to be clarified. In the main text, when we discussed 220 phased genome assemblies, we meant haploid assemblies. We now make this clearer stating there were 110

phased genomes, which accounts for 220 haplotypes. See the previous response for the suggested changes in the main text.

In addition, we provide a summary of assembly completeness in supplementary figure 3. Assembly completeness for HPRC (release 1) assemblies, which have been assembled using HiFi reads only, is 41% while the latest HGSC assemblies, completed using both HiFi and ONT reads, show 76% completeness at 22q11.2 region. Overall, 61.4% (135 out of 220) of 22q11.2 haplotypes passed our assembly quality criteria. See the detailed description of our assembly validation steps in the following answer.

This is a very critical issue in the accurate assembly of SD/LCR containing regions. Unless both haplotypes from a given individual are assembled into single contigs without any unassigned sub-contigs, it is difficult to assess the accuracy of the assembly.

We agree with the reviewer that the assembly quality is a critical point in our subsequent analysis and interpretation of the results. We thoroughly validated all our assemblies by mapping sample-specific long high-fidelity sequence reads (HiFi, PacBio) to each assembly and used the NucFreq tool (Vollger et al. 2019) to confirm the accuracy of all assembled haplotypes that span the 22q11.2 region in a single continuous contig. Phasing accuracy of all our assembled haplotypes was ensured by using trio-based phasing in HPRC1 (release 1) (Liao et al. 2023) assemblies and using Strand-seq phasing information in HGSC (Logsdon et al. 2025) assemblies.

LCRA is structurally the most diverse region of 22q11.2 and thus the most difficult to assemble. Because of this we argue that if the same LCRA structure is assembled independently in two or more assemblies this can serve as an extra layer of validation. We compiled a detailed supplementary note where we specify, in detail, all steps taken to validate our assemblies and the proper structure of the most complex LCRA (see below).

Supplementary Note 1:

Evaluation of 22q11.2 assembly accuracy of human diversity panel. To ensure the quality of assembled haplotypes analyzed in this study, we set to evaluate the assembly accuracy using multiple orthogonal methods and datasets.

- 1) From the total of 220 haplotypes, we selected only those haplotypes with a region corresponding to the 22q11.2 region (chr22:18000000-23000000, T2T-CHM13 coordinates) assembled in a single continuous contig with no N's within a sequence that would point to scaffolded assembly gaps.
- 2) We re-aligned PacBio HiFi reads back to each assembly using minimap2 (**Methods**) in order to evaluate each haplotype assembly for the presence of assembly collapses in highly identical LCR regions (especially in LCRA and D) (Vollger et al. 2019). The presence of an increased coverage over a defined region along with an increased count of the second most abundant allele is indicative of a collapsed assembly of a piece of

DNA present in a given region in more than one copy (**Supplementary Fig. 47, Supplementary Table 8**).

- 3) To further evaluate the accuracy of HPRC (release 1) assemblies generated using HiFi reads only (Liao et al. 2023), we obtained access to HPRC2 (release 2) assemblies completed using both HiFi and ONT reads. For a subset of these samples, we were able to compare assembly accuracy for 35 haplotypes. Of the 35 haplotypes, there are 30 for which the sequence identity between release 1 and release 2 HPRC assemblies is more than 99.99% (**Supplementary Fig. 48**). We noticed, however, five haplotypes with a lower sequence accuracy (>99.5% but <99.99%) due to an inversion in a distal part of the LCRD. We predict this is a misassembly within HPRC (release 1) assemblies, so we now report affected haplotypes in **Supplementary Table 8**. We do still use these five haplotypes in our subsequent analysis as these low-frequency discrepancies have no effect on our prediction of deletion predisposition at 22q11.2 because of the location of the putative misassembly.
- 4) As another metric of accuracy, we searched for biological replication—i.e., evidence that the same structure had been observed in two independent human genomes. Specifically, we took advantage of available HPRC2 (release 2) data (**Data Availability**) to evaluate the accuracy of the higher-order structure of LCRA among all singleton haplotypes (n=47) defined among a total of 63 unique LCRA structures. For each singleton structure we searched the 418 HPRC2 structures of LCRA in order to find the best matching one. We did this by encoding each repeat unit by a unique number and comparing them against all HPRC2 structures. We calculated the distance between these numeric encodings of each structure using R package stringdist and its function 'seq_dist' (function parameter: method='osa'). We consider a singleton structure to be validated by HPRC2 if there is a matching HPRC2 structure with less ≤5% divergence. Based on this threshold, we confirm 23 singleton structures are seen at least once in the HPRC2 dataset (**Supplementary Fig. 49**).
- 5) Lastly, we compared the organization of LCRA to previous fiber-FISH and optical-mapping-based analyses of the same samples where applicable. To validate the assemblies, the structure of LCRA among the 133 haplotypes has been investigated using the sequence of the probes previously used to characterize the region by fiber-FISH and optical genome mapping (Demaerel et al. 2019). The probe content of each haplotype was inferred using BLAST and selecting matches with at least 87% identity compared to the fiber-FISH probe and with length of at least 50% of the fiber-FISH probe. LCRAs are represented using the repeat units defined by Demaerel et al. (2019). Most assemblies have a structure that matches what has been previously described. Haplogroups 13* and 14 differ slightly from previously described structures as one probe is only partially included (42% of the probe length is included in the assembly) (**Supplementary Fig. 50**). A detailed comparison of the structures is summarized in **Supplementary Table 9**. Similarly, we also validated the assembly-based structure of LCRA and D using fiber-FISH for three family duos (AD009, AD010, and AD013) (Demaerel et al. 2019). All family duos are in line with the fiber-FISH observed LCRA and D structure (**Supplementary Fig. 51**). The structural validity of the Coriell trio was confirmed using local reassembly with NOVOLOci (**Methods**). We, however, observe a

putative haplotype switch error downstream from LCRD, which has no effect on the proper mapping of the 22q11.2DS breakpoint (**Supplementary Fig. 44**).

We note that the chromosome-scale phasing accuracy of all our assembled haplotypes was ensured by using trio-based phasing in HPRC (release 1) (Liao et al. 2023) assemblies or using Strand-seq phasing information in HGSV (Logsdon et al. 2025) assemblies.

Supplementary Figure 47: Phased assembly validation using NucFreq.

Two examples of read-coverage profiles of aligned PacBio HiFi reads to the two phased genome assemblies of the 22q11.2 region (chr22:1800000-23000000 in T2T-CHM13 coordinates). **a**) A visualization of the most abundant base at each position of HG04036-H2 is shown in black. The second most abundant base is shown in red. Extended regions with observable increased coverage for the most abundant base ('Max') often coincide with increased frequency of the second ('Remainder') most abundant base (see arrow). Such regions are indicative of a collapsed assembly where not all copies of paralogous duplications are fully resolved. Misassemblies would be represented as an absence of sequence reads mapping across a region. **b**) This visualization shows the same frequency of the most and second most abundant bases along the haplotype from sample NA19705-H1. In this case we consider this assembly to be complete and correct as there are no extended regions of increased frequency of the secondary base ('Remainder' base).

Supplementary Figure 48: Comparison of HPRC1 and HPRC2 assemblies for the same haplotypes.

a) A barplot showing the percent agreement between HPRC1 (release 1) and HPRC2 (release 2) assemblies generated using PacBio HiFi reads only for HPRC1 and both HiFi and ONT reads for HPRC2 assemblies. Haplotypes marked with an asterisk are those where we observed a distal inversion within LCRD (see example in panel b). **b**) Visualization of binned (bin size: 10 kbp) sequence identity between the HPRC2 assembly for samples HG01071-H2 aligned to the HPRC1 assembly of the same haplotype. Alignments are colored by sequence identity. On top of the HPRC1 assembly there is a DupMasker annotation shown as colored arrowheads. Continuous LCRA and D regions are also highlighted. There is also a structural variant (SV, ≥ 50 bp) colored by blue and red outlines for insertions (INS) and deletions (DEL), respectively. Lastly, we mark an inverted region between the HPRC1 and HPRC2 assemblies visible in the distal region of LCRD. ROI - region of interest (T2T-CHM13:18-23Mbp).

Supplementary Figure 49: Validation of singleton LCRA structures using HPRC2 (release 2) assemblies.

Singleton LCRA structures (n=47) observed among 133 noninverted haplotypes are presented in each row (always on the top). Then, for each singleton haplotype, we present the closest haplotype structure observed among HPRC2 assemblies (always at the bottom). Each haplotype structure is presented as directional arrowheads colored by a SD repeat unit (120 kbp, 105 kbp, 25 kbp, and 28 kbp; see legend; 41 kbp repeat unit not shown for simplicity). Singleton haplotypes whose exact structure was observed among HPRC2 samples are marked by green points.

Supplementary Figure 50: Representation of LCRA assemblies according to fiber-FISH duplcons.

Each haplotype has been decomposed in duplcons based on its fiber-FISH probe composition. The yellow, cyan, magenta, blue, green and white duplcons are 25 kbp, 29 kbp, 155 kbp, 100 kbp, 80 kbp and 31 kbp, respectively (Demaerel et al. 2019). LCRA structures defined by fiber-FISH duplcons and assembly are very similar in content, size, and orientation. The corresponding assembly haplogroup is given on the right. The assembly-based SD repeat units, defined in this manuscript, blue (25 kbp), yellow (28 kbp), light red (105 kbp) and brown (120 kbp) match the white (31 kbp), cyan (29 kbp), blue (100 kbp) and magenta (155 kbp) (see the legend). The fiber-FISH-based duplcon with blue stripes matches the 41 kbp duplcon defined within our assemblies (Fig. 2c).

Supplementary Figure 51: Validation of 22q11.2DS families using fiber-FISH.

LCRA and LCRD alleles of proband and parent of origin are represented using the SD repeat units as published by Demareel et al. (2019). The region in which the breakpoint occurred is shown for each duo in a black box. The assembled haplotypes have been decomposed in duplicons based on the fiber-FISH probe composition. All haplotypes matched the fiber-FISH defined structure.

3) This raises further concerns about the results and conclusions presented here which were derived from the assembled haplotypes.

Notably, NAHR is likely to be affected by the SD/LCR structure on both haplotypes in the Parent of Origin. Thus, any conclusions about African ancestry genomes (versus other ancestries) being more predisposed to one type of rearrangement versus another based on single haplotypes from any individual seems premature.

The reviewer raises an important point regarding the predisposition analysis that resulted in a major revision to the paper (although it did not, surprisingly, alter the major conclusions). Because most chromosome 22q11.2 microdeletions arise from “trans” exchange events between homologous chromosomes (as opposed to sister chromatids or intrachromosomal), we included an analysis of chromosome 22 homologues from individuals where both chromosome 22q11.2 regions were fully resolved. For example, it was previously reported that the frequency of intra (cis) versus interchromosomal (trans) exchanges leading to 22q11.2DS is in favor of interchromosomal ones (19 inter vs. 1 intrachromosomal, 95% vs. 5 %) (Saitta et al. 2004).

To make sure our predictions regarding the deletion predisposition at 22q11.2 are robust, we expanded the analysis to include samples from both the HGVC (n=65) and HPRC2 (release 2; n=232) in order to find as many samples as possible with both parental haplotypes fully and accurately assembled. In total, among the HGVC and HPRC2 assemblies we identified 233 (41 from HGVC and 192 from HPRC2) samples where both parental haplotypes are assembled in a single continuous contig.

In this new analysis we tested all possible intrachromosomal (cis) and interchromosomal (trans) interactions between all copies of the 105 kbp repeat unit present in LCRA or LCRD that could be implicated in 22q11.2 deletion formation via NAHR. We find that in both cis and trans interactions of the 105 kbp repeat unit between LCRA and D, the reverse-oriented repeats have slightly higher sequence identity than those that are forward oriented, represented by a median sequence identity 99.1% and 99%, respectively (**Supplementary Fig. 42**). Given this observation, we report in our paper the predisposition to 22q11.2DS based on interchromosomal interactions between LCRA and D and revised the main figure accordingly (see **Figure 5** below).

Based on these new analyses involving both homologs of chromosome 22, we again observe African genomes are significantly (Fisher's exact test, two-sided, Bonferroni corrected, $p = 1.14 \times 10^{-6}$, odds ratio = 25.8) less predisposed to 22q11.2DS than other ancestries. In addition to this, we report a significantly (Fisher's exact test, two-sided, Bonferroni corrected, $p = 1.19 \times 10^{-7}$, odds ratio = 0) higher predisposition to 22q11.2DS among individuals of East Asian ancestry although there is as of yet no epidemiological data to support this observation. We argue that these observations can be explained by the unique organization of the 105 kbp repeat unit in LCRA and D. Among haplotypes that carry exactly one copy of the 105 kbp repeat unit in LCRA and D, we see that East Asians have often 105 kbp repeats in forward orientation while among Africans (and also Europeans) these are often in reverse orientation promoting formation of recurrent inversions as opposed to copy number changes associated with the microdeletion.

We modified Extended Data Figure 1 and Figure 5 in the main paper to include these new results (see below):

Supplementary Figure 42: Sequence identity of the 105 kbp repeat pairs between LCRA and D.

The boxplot shows the overall sequence identity for all intra (cis) and interchromosomal (trans) pairs of the 105 kbp repeat unit between LCRA and D. Overall reverse-oriented (gray) LCRA and D pairs have median identity 99.1% while those in forward orientation (dark red) have median identity 99%. Boxes represent interquartile range (IQR), including median line; whiskers extend to 25% - 1.5 x IQR and 75% + 1.5 x IQR; outliers are shown as black dots.

Figure 5: 22q11.2DS predisposition and microdeletion rearrangements.

a) Pictogram of intrachromosomal (cis) and interchromosomal (trans) interactions between the 105 kbp repeat unit within LCRA and LCRD. Forward- and reverse-oriented 105 kbp repeat units between LCRA and D are shown in dark red and gray, respectively. **b)** Counts of all possible LCRA and D interactions of the 105 kbp repeat unit per sample (n=233, plus four 22q11.2DS transmitting parents). Only interchromosomal (trans) interactions are evaluated (see Extended Data Fig. 1 for intrachromosomal interactions). Direct (forward)-oriented pairs are colored in dark red and inverted (reverse) pairs are shown in gray. At the base of the plot we show the ancestry origin of each haplotype. Black arrows mark the results for four transmitting parents. **c)** Evaluation of the proportions of each ancestry (AFR - African, AMR - American, EUR - European, EAS - East Asian, and SAS - South-East Asian) among haplotypes defined as either predisposed or protected against 22q11.2DS with respect to the rest of the ancestries ('Other' - gray). We used Fisher's exact test (two-sided) to test the significance of the difference between these proportions. Above each barplot is the reported p-value after Bonferroni correction. **d)** Visualization of binned (bin size: 10 kbp) sequence identity between probands with respect to both haplotypes (H1 and H2) of a transmitting parent for four families. The proband haplotype is in the middle while parental haplotypes H1 and H2 are at the top and bottom, respectively. Alignments are colored by sequence identity and likely inherited portions of parental haplotypes in the proband are highlighted by fully colored alignments (darker shades of red) while others are shown with 50% transparency. On top of each haplotype are HOR annotations depicted as arrowheads colored by SD repeat units. Refined deletion breakpoints within the 105

kbp repeat units are marked by black arrowheads. **e)** Coverage summary of the longest continuous region of high identity ($\geq 99.5\%$) among all alignments between all possible interchromosomal interactions of 105 kbp long repeat units per sample. We distinguish the coverage for the direct (forward, dark red) and inverted (reverse, gray) alignments. We mark the position of putative deletion breakpoints in four probands as black arrowheads.

Extended Data Figure 1: Predisposition to 22q11.2DS via intrachromosomal NAHR.

a) Model of intra- and interchromosomal non-allelic homologous recombination (NAHR) and its possible outcomes depending on the relative orientation of segmental duplications (or low copy repeats). **b)** Counts of all possible LCRA and D interactions of the 105 kbp repeat unit per single haplotype (n=134, including T2T-CHM13 haplotype). Only intrachromosomal (cis) interactions are evaluated. Direct (forward)-oriented pairs are colored in dark red and inverted (reverse) pairs are shown in gray. At the base of the plot we show the ancestry origin of each haplotype. **c)** Shows evaluation of the proportions of each ancestry (AFR - African, AMR - American, EUR - European, EAS - East Asian, and SAS - South-East Asian) among haplotypes defined as either predisposed or protected against 22q11.2DS with respect to the rest of the ancestries ('Other' - gray). We used Fisher's exact test (two-sided) to test the significance of the difference between these proportions. Above each barplot is a reported p-value after Bonferroni correction.

Thus, the entire Result section titled "Predisposition to 22q11.2DS..." on Page 12 as well as any resulting conclusions in the Discussion need to be reconsidered and restated based on this limitation of their assemblies and data used.

We revised the Results section based on the updated analysis that includes HPRC2 (release 2) assemblies (and subsequently include them as a banner author in the paper).

Revised Results section:

Predisposition to 22q11.2DS predicted to differ by ancestry. Because the majority of chromosome 22q11.2 recurrent microdeletions arise as a consequence of NAHR between LCRA and LCRD driven by the 105 kbp SD repeat unit (Vervoort et al. 2025), we examined the orientation and percent identity of this repeat unit among 133 sequence-resolved haplotypes (**Extended Data Fig. 1a**). We find that about a third of all haplotypes (n=43) carry exactly one copy of the 105 kbp repeat unit in both LCRA and D, differing only by an inversion of the 105 kbp repeat unit in LCRA. This renders the 105 kbp repeat units in either a direct (n=31) or an indirect (n=12) orientation between LCRA and D (**Supplementary Fig. 40**) making them, theoretically, either predisposed or protected for A-to-D deletion, respectively. We extended this analysis to all 133 human haplotypes and tracked the orientation of all pairs of the 105 kbp repeat unit mapping between LCRA and D (**Methods**). We classified and rank ordered 22q11.2 haplotypes as either predisposed to LCRA-to-D inversion or A-to-D deletion (**Extended Data Fig. 1b, Methods**) under the assumption that more SD pairs of directly orientated 105 kbp repeat units would favor deletion formation while those in reverse orientation would favor inversion formation (Cooper et al. 2011; Inoue and Lupski 2002). We find significantly more haplotypes of African ancestry are predicted as being protected against 22q11.2DS than those of non-African ancestry (Fisher's exact test, two-sided, Bonferroni corrected, p=0.008, odds ratio: 4.5) (**Extended Data Fig. 1c**).

However, this predisposition analysis only considers intrachromosomal (cis) NAHR within a single human haplotype. It has been shown that NAHR at 22q11.2 arises more frequently (in about 95% cases) as interchromosomal (trans) NAHR events between maternal and paternal haplotypes during meiosis (Saitta et al. 2004) (**Extended Data Fig. 1a and Fig. 5a**). To assess this we identified genomes where both haplotypes were fully sequenced and assembled. In total, we identified 233 samples among HGSVC and HPRC2 (release 2) assemblies with both haplotypes fully assembled and compared the sequence identity and relative orientation of all possible interchromosomal pairs of the 105 kbp repeat unit between LCRA and D (**Fig. 5a, Methods, Data Availability**). We find the sequence identity of both cis and trans interactions, between the 105 kbp units within LCRA and D to be >99% with the reverse-oriented SDs having a slightly higher sequence identity (0.1% increase) than those in forward orientation (**Supplementary Fig. 42**). In total, we predict 37 protected and 39 predisposed samples given the interchromosomal orientation of LCRA and D either in reverse or forward orientation, respectively (**Fig. 5b**). Again, in line with intrachromosomal analyses, African genomes are predicted to be significantly less predisposed (Fisher's exact test, two-sided, Bonferroni corrected, $p = 1.14 \times 10^{-6}$, odds ratio = 25.8) to 22q11.2DS consistent with the >3-fold reduced prevalence reported for African Americans epidemiologically (McDonald-McGinn et al. 2005;

Pastor et al. 2025). In addition, we predict that individuals of East Asian ancestry are significantly enriched (Fisher's exact test, two-sided, Bonferroni corrected, $p = 1.19 \times 10^{-7}$, odds ratio = 0) for predisposed haplotypes, although there is no epidemiological data at present to support this genomic prediction.

Sequence breakpoint characterization of 22q11.2DS patients. To directly test the relationship between LCRA and D structures and sporadic chromosome 22q11.2 deletions associated with disease, we sequenced and assembled the 22q11.2 region from four families where a *de novo* deletion had occurred in a child with developmental disabilities. Because the transmitting parent was already known in three cases (Vervoort et al. 2025), we sequenced and assembled parent–child duos while for the fourth we considered all three members of the parent–child trio (**Data Availability**). Using LRS and assembly approaches, we fully assembled and performed QC analysis on the chromosome 22q11.2 region for each transmitting parent and a child (proband) (**Supplementary Note 1**). We compared the phased assembly of the proband (with A-to-D deletion) with both haplotypes (H1 and H2) of the transmitting parent to define at the sequence level the breakpoint of the rearrangement. Given the accuracy of the assembly, near-perfect sequence identity could be used to readily track parental haplotypes against the child's assembled chromosome (**Fig. 5d**, as we demonstrated previously (Porubsky et al. 2025)), in order to define recombination breakpoints.

For family 1 (AD009), a single breakpoint region was readily identified deleting ~2.5 Mbp from H1, as a result of an unequal crossover between H1 and H2 likely when the quadrivalent was formed during meiosis I allowing for interchromosomal exchange between homologs. In this family, the deletion breakpoint occurs within the 105 kbp repeat unit (**Fig. 5e**) and we refine the breakpoint to a very narrow range of 100 bp (**Fig. 5c, Supplementary Fig. 43, Methods**) mapping nearby an exon of the transcribed *GGT2* gene family. In family 2 (AD010) and family 3 (AD013), we observe a nearly identical pattern of haplotype exchange between parental homologs as in family 1, consistent with unequal crossover within the 105 kbp repeat. Here, we narrowed down the deletion breakpoint to a 200 bp and 2,303 bp region for family 2 and 3, respectively (**Supplementary Fig. 43**). Last, in family 4 (the publicly available parent–child trio; **Data Availability**), we again see the expected interchromosomal exchange between parental homologs at the deletion breakpoint mapped within 1,150 bp range. In addition, we observe a large-scale insertion (40.8 kbp) at the deletion (**Supplementary Fig. 44a-c**). Taken together, all four families have deletion breakpoints mapping to the 105 kbp SD repeat unit albeit at different locations (**Fig. 5e**). These breakpoints can be found within or in the vicinity of the high-identity regions ($\geq 99.5\%$) clustered in two locations (around 50 kbp and 74 kbp) of the 105 kbp repeat unit (**Fig. 5e, Supplementary Table 4**). While this association with the larger 105 kbp repeat unit was predicted before (Demaerel et al. 2019; Zhou et al. 2024), we now provide fully resolved haplotypes with base-pair resolution.

Finally, we projected all haplotypes of AD-deletion transmitting parents ($n=4$) against our population genetic analysis of fully assembled haplotypes from 233 samples where we had initially predicted predisposition to or protection against the 22q11.2DS based on the orientation and sequence identity of interchromosomal interactions between LCRA and D (**Fig. 5b**). When

rank ordering parental haplotypes on our scale of AD-deletion predisposition, we find two out of four parents are predicted to be fully predisposed to AD-deletion. One transmitting parent ranks as equally predisposed and protected while the last parent ranks as more protected suggesting that the sequence structure offers predictive power but is not absolutely deterministic of rearrangement susceptibility as long as there are still SDs in a forward orientation that can predispose to NAHR (**Fig. 5e**).

Revised discussion points:

Importantly, these expansions do not always correlate with a predicted increased probability for NAHR. Considering percent identity and repeat orientation between LCRA and D, an analysis of both interchromosomal and intrachromosomal configurations predicts that individuals of African ancestry are more likely to be predisposed to inversions as opposed to large-scale *de novo* deletions (**Extended Data Fig. 1c, Fig. 5c**). Thus, despite the larger size of LCRA and higher copy number of the 105 kbp SD repeat unit, Africans are more likely to be fully protected against 22q11.2DS than those of non-African ancestry because of the preponderance of inversely oriented repeats between the LCR blocks. This sequence structure prediction matches closely an epidemiological observation made 20 years ago where 22q11.2DS deletions were found to be three times less common among patients of African-American ancestry (McDonald-McGinn et al. 2005; Pastor et al. 2025). This reduced prevalence was subsequently shown not to be the result of phenotypic ascertainment biases and was hypothesized to be the result of differences in the LCR structures (McDonald-McGinn et al. 2023). In contrast, we find that the LCR structures of individuals of East Asian ancestry are significantly (Fisher's exact test, two-sided, Bonferroni corrected, $p = 1.19 \times 10^{-7}$, odds ratio = 0) predisposed to interchromosomal NAHR due their predominant direct orientation of the 105 kbp repeat in between LCRA and D; however, we were not able to find clinical results to support this observation. Nevertheless, these findings suggest that it might be more important to screen individuals of East Asian descent (i.e., Japan) for 22q11.2DS given their particular structural configurations that predispose to NAHR (**Supplementary Fig. 46**).

Supplementary Figure 46: Population labels of samples predicted to be 22q11.2DS susceptible.

The height of each bar shows the number of samples marked as predisposed to 22q11.2DS colored by population labels obtained from the 1KG project. The three letter abbreviations are 1KG-specific population labels (x-axis). Shades of green mark individuals of East-Asian ancestry (EAS).

4) Similarly, it is difficult to assess the accuracy of the assemblies in the patient family trios as these data are currently not available. It would be useful for the reader to understand how the authors were able to assemble these haplotypes with confidence when the T2T consortium genomes appear to be partially (or entirely) incompletely assembled across the 22q11 region.

As with all other assemblies, one of the most reliable strategies was to remap PacBio HiFi reads from each sample to test if they support each assembled haplotype. However, this analysis can only detect collapses in our assemblies and is not informative for detecting haplotype switches. To account for this we also compared our patient assemblies to the fiber-FISH haplotypes generated by Demaerel et al. (2019) (see **Supplementary Note 1**). All assemblies matched the fiber-FISH except for the AD009 father for which a switch was observed in the first assembly (see below). Therefore, a targeted assembly of this clinical sample was performed using NOVOloci in order to correct the error (**Methods**). We also performed targeted assembly for other samples where we suspected an assembly error (for instance Coriell trio).

Below is an example of a clinical sample (AD009-father) locally reassembled using the NOVOloci tool based on ONT reads only. We can now see that the previously observed double recombination event is no longer predicted; rather, we see a simple recombination event at the deletion breakpoint, which is in line with the expected interchromosomal NAHR.

Targeted assembly of 22q11.2DS families (Methods)

Hifiasm assemblies were validated by performing targeted assemblies of the 22q11.2 region with NOVOloci (v0.5)(Dierckxsens et al. 2025). We applied targeted assembly to each family with observable phasing errors (AD009, Coriell trio). We assembled the region twice, using one proximal and one distal seed to increase confidence. As the proximal seed, we selected a 500

bp sequence from the CHM13 reference genome starting at position 18,000,000; the distal seed was a 500 bp sequence starting at position 23,800,000. These NOVOLOci assemblies were generated using only ONT reads and therefore had a lower QV. Consequently, they were used exclusively to investigate suspected misassemblies in the hifiasm assemblies.

Revision figure:

Top: Incorrectly phased parental haplotypes with a large-scale haplotype phasing error.
 Bottom: Correctly reassembled parental haplotypes with a single haplotype switch at the 22q.11.2 deletion breakpoint.

5) The authors may have inadvertently forgotten to acknowledge that the localization of deletion breakpoints (Page 14) within what they call the 105 Kb repeat unit was originally shown by Demaerel et al (Ref 25), who refer to that region as SD22-4. Granted, that the current work provides sequence level resolution to confirm the findings of Demaerel et al. and others.

We apologize for this oversight on our side. We now add the reference to previous work when talking about the breakpoint resolution.

Revised (in bold):

These breakpoints can be found within or in the vicinity of the high-identity regions ($\geq 99.5\%$) clustered in two locations (around 50 kbp and 74 kbp) of the 105 kbp repeat unit (**Fig. 5e**). **While this association with the larger 105 kbp repeat unit was shown before** (Demaerel et al. 2019; Zhou et al. 2024), **we now provide fully resolved haplotypes with base-pair resolution.**

6) The authors seem to downplay the previous work of others when it is clear that many of their sequence resolved results were already predicted or known based on previous studies. This includes a large body of work on PATRRs in 22q11 rearrangements, and breakpoints localized in the SDs/LCRs on 22q11. Proper acknowledgement of these studies would be beneficial and help the reader understand the progression of the work on this difficult region of the genome and the authors' contribution to the overall deciphering of the complexity of 22q11.2.

This omission was not intended as we were focused on fully sequence-resolved structures, which are often difficult to compare against optical mapping or FISH data that offer lower resolution. We spent the last few weeks comparing our findings with earlier investigations and now include more references to previous work (see below).

Revised #1:

Previous studies using fluorescence *in situ* hybridization (FISH) and optical mapping (Shimajima et al. 2011; Guo et al. 2016; Vervoort and Vermeesch 2022) provided valuable insights into the structural diversity of this locus, but these methods lack the resolution necessary to precisely map deletion breakpoints or explore the fine-scale diversity of the underlying LCRs flanking this region.

Revised #2:

More recently, optical mapping methods were coupled with ultra-long Oxford Nanopore Technologies (UL-ONT) sequencing approaches to locally assemble and characterize breakpoints in 22q11.2DS families (Pastor et al. 2020; Vervoort et al. 2025; Zhou et al. 2024). While this methodology helped refine recombinant breakpoint regions, standard assembly methods failed to fully resolve the chromosome 22q11.2 region or the complexity of underlying rearrangements in either patients or population controls.

Revised #3:

At the boundaries of 25 and 28 kbp long HORs, we locate the previously defined pockets of palindromic AT-rich repeats (PATRRs) (Kurahashi et al. 2007; Kato et al. 2012; Correll-Tash et al. 2021).

Minor Issue:

1) There is some gibberish on line 5, page 7.

We apologize for this though believe it may have been a system conversion problem as we cannot locate this issue, even though called out by two reviewers. However, we have carefully checked our manuscript for resubmission.

Reviewer #2 (Remarks to the Author):

The manuscript by Coughlin et al. presents a comprehensive study of the structural haplotype architecture in the 22q11.2 region, combining PacBio HiFi long-read data from diverse global populations. The authors link specific structural haplotypes with predisposition to recurrent rearrangements, specifically, LCR22A–D deletions and inversions, and find striking population differences. This is an elegant and ambitious study with substantial implications for both genome biology and clinical genetics. However, several aspects of figure presentation, data summarization, and quality control reporting could be improved to enhance clarity and reproducibility.

We thank the reviewer for acknowledging the value of this study to the scientific community and for all useful comments to improve this manuscript.

Major Points

1. Figure design and visual clarity

Figures contain a large number of data points, tracks, or labels crammed into a small space. While they are information-rich and visually appealing, many of them are too small, overly dense, or lack clear labeling, which significantly reduces their accessibility. The font size of labels is consistently too small.

Suggestion: Increase font size and streamline visual elements. Consider separating complex panels into multiple figures or using simplified summary schematics alongside detailed plots. A summary table or flowchart explaining the key structural haplotypes (such as SD22-3 and SD22-4 configurations) and their predisposition to NAHR-mediated events across populations would help guide the reader through the results.

We simplified the number of figure panels and the presentation of the Results. We also increased the font size across all figures to make sure all information is easily visible and accessible to the reader. See below for an example of simplified Figure 2 where we replaced the duplication structures reported in panel C with higher-order structures to make the differences in LCRA more obvious. We also removed some nonessential figure panels and moved them to the supplement in order to improve readability of our main results.

Figure 2: Chromosome 22q11.2 structural diversity and duplcon higher-order structure. a) Violin plot showing length distribution for LCRA-D along with mean LCR block size (red dots). b) Size-sorted distribution of LCRA size (colored by cluster ID) with the mean size of each cluster (horizontal dotted line) and the mean size difference between each step indicated by the number. c) Higher-order repeat (HOR) structure of 63 distinct LCRA haplogroups (colored arrowheads show size and orientation of five repeat units 25, 28, 41, 105, and 120 kbp in length). Note that the 41 kbp repeat unit maps within the 105 kbp repeat unit and is only rarely seen separately. Stacked barplot shows the frequency of each haplogroup among continental population groups. Inset: Shows a significant size difference (Wilcoxon rank-sum test, two-sided, $p=0.0047$) of LCRA among haplotypes of African and non-African ancestry.

2. Missing details on quality control and data filtering

The resolution and phasing presented in the study are impressive, but the QC thresholds and filtering steps used to achieve this level of accuracy are not clearly reported.

For instance, what coverage was required? What were the minimum read lengths? How many of the initial samples failed assembly or phasing, and why?

These details are important for readers hoping to replicate or build on this work.

Thank you for the comment. We now present a detailed summary, including the specific steps we took to filter out low-quality assemblies in the **Supplementary Notes** (see below). We also describe what methods and approaches we used to make sure our selected assemblies meet quality requirements for subsequent analysis. The overall sequence agreement for 35 tested samples, assembled with both HiFi reads only as well as a combination of HiFi and ONT reads, is $>99.99\%$ (**Supplementary Note 1**). Please note that assembly quality metrics per sample, such as read N50 coverage or contig lengths, are presented in the original papers from where the assemblies were obtained, thus we decided to not to replicate these results here. We, however, present such values for long-read datasets and assemblies generated as part of this

study. Namely two samples (HG01888 and NA19315) that carry an LCRA-D inversion, including three clinical family duos and one family trio with probands carrying LCRA-D deletion.

Assembly and long-read data statistics generated in this study are presented in **Supplementary Tables 5 and 6**.

Supplementary Note 1:

Evaluation of 22q11.2 assembly accuracy of human diversity panel. To ensure the quality of assembled haplotypes analyzed in this study, we set to evaluate the assembly accuracy using multiple orthogonal methods and datasets.

- 1) From the total of 220 haplotypes, we selected only those haplotypes with a region corresponding to the 22q11.2 region (chr22:18000000-23000000, T2T-CHM13 coordinates) assembled in a single continuous contig with no N's within a sequence that would point to scaffolded assembly gaps.
- 2) We re-aligned PacBio HiFi reads back to each assembly using minimap2 (**Methods**) in order to evaluate each haplotype assembly for the presence of assembly collapses in highly identical LCR regions (especially in LCRA and D) (Vollger et al. 2019). The presence of an increased coverage over a defined region along with an increased count of the second most abundant allele is indicative of a collapsed assembly of a piece of DNA present in a given region in more than one copy (**Supplementary Fig. 47, Supplementary Table 8**).
- 3) To further evaluate the accuracy of HPRC (release 1) assemblies generated using HiFi reads only (Liao et al. 2023), we obtained access to HPRC2 (release 2) assemblies completed using both HiFi and ONT reads. For a subset of these samples, we were able to compare assembly accuracy for 35 haplotypes. Of the 35 haplotypes, there are 30 for which the sequence identity between release 1 and release 2 HPRC assemblies is more than 99.99% (**Supplementary Fig. 48**). We noticed, however, five haplotypes with a lower sequence accuracy (>99.5% but <99.99%) due to an inversion in a distal part of the LCRD. We predict this is a misassembly within HPRC (release 1) assemblies, so we now report affected haplotypes in **Supplementary Table 8**. We do still use these five haplotypes in our subsequent analysis as these low-frequency discrepancies have no effect on our prediction of deletion predisposition at 22q11.2 because of the location of the putative misassembly.
- 4) As another metric of accuracy, we searched for biological replication—i.e., evidence that the same structure had been observed in two independent human genomes. Specifically, we took advantage of available HPRC2 (release 2) data (**Data Availability**) to evaluate the accuracy of the higher-order structure of LCRA among all singleton haplotypes (n=47) defined among a total of 63 unique LCRA structures. For each singleton structure we searched the 418 HPRC2 structures of LCRA in order to find the best matching one. We did this by encoding each repeat unit by a unique number and comparing them against all HPRC2 structures. We calculated the distance between these numeric encodings of each structure using R package stringdist and its function 'seq_dist' (function parameter: method='osa'). We consider a singleton structure to be

validated by HPRC2 if there is a matching HPRC2 structure with less $\leq 5\%$ divergence. Based on this threshold, we confirm 23 singleton structures are seen at least once in the HPRC2 dataset (**Supplementary Fig. 49**).

- 5) Lastly, we compared the organization of LCRA to previous fiber-FISH and optical-mapping-based analyses of the same samples where applicable. To validate the assemblies, the structure of LCRA among the 133 haplotypes has been investigated using the sequence of the probes previously used to characterize the region by fiber-FISH and optical genome mapping (Demaerel et al. 2019). The probe content of each haplotype was inferred using BLAST and selecting matches with at least 87% identity compared to the fiber-FISH probe and with length of at least 50% of the fiber-FISH probe. LCRAs are represented using the repeat units defined by Demaerel et al. (2019). Most assemblies have a structure that matches what has been previously described. Haplogroups 13* and 14 differ slightly from previously described structures as one probe is only partially included (42% of the probe length is included in the assembly) (**Supplementary Fig. 50**). A detailed comparison of the structures is summarized in **Supplementary Table 9**. Similarly, we also validated the assembly-based structure of LCRA and D using fiber-FISH for three family duos (AD009, AD010, and AD013) (Demaerel et al. 2019). All family duos are in line with the fiber-FISH observed LCRA and D structure (**Supplementary Fig. 51**). The structural validity of the Coriell trio was confirmed using local reassembly with NOVOLocI (**Methods**). We, however, observe a putative haplotype switch error downstream from LCRD, which has no effect on the proper mapping of the 22q11.2DS breakpoint (**Supplementary Fig. 44**).

We note that the chromosome-scale phasing accuracy of all our assembled haplotypes was ensured by using trio-based phasing in HPRC (release 1) (Liao et al. 2023) assemblies or using Strand-seq phasing information in HGSC (Logsdon et al. 2025) assemblies.

Supplementary Figure 47: Phased assembly validation using NucFreq.

Two examples of read-coverage profiles of aligned PacBio HiFi reads to the two phased genome assemblies of the 22q11.2 region (chr22:1800000-23000000 in T2T-CHM13 coordinates). **a**) A visualization of the most abundant base at each position of HG04036-H2 is shown in black. The second most abundant base is shown in red. Extended regions with observable increased coverage for the most abundant base ('Max') often coincide with increased frequency of the second ('Remainder') most abundant base (see arrow). Such regions are indicative of a collapsed assembly where not all copies of paralogous duplications are fully resolved. Misassemblies would be represented as an absence of sequence reads mapping across a region. **b**) This visualization shows the same frequency of the most and second most abundant bases along the haplotype from sample NA19705-H1. In this case we consider this assembly to be complete and correct as there are no extended regions of increased frequency of the secondary base ('Remainder' base).

Supplementary Figure 48: Comparison of HPRC1 and HPRC2 assemblies for the same haplotypes.

a) A barplot showing the percent agreement between HPRC1 (release 1) and HPRC2 (release 2) assemblies generated using PacBio HiFi reads only for HPRC1 and both HiFi and ONT reads for HPRC2 assemblies. Haplotypes marked with an asterisk are those where we observed a distal inversion within LCRD (see example in panel b). **b)** Visualization of binned (bin size: 10 kbp) sequence identity between the HPRC2 assembly for samples HG01071-H2 aligned to the HPRC1 assembly of the same haplotype. Alignments are colored by sequence identity. On top of the HPRC1 assembly there is a DupMasker annotation shown as colored arrowheads. Continuous LCRA and D regions are also highlighted. There is also a structural variant (SV, ≥ 50 bp) colored by blue and red outlines for insertions (INS) and deletions (DEL), respectively. Lastly, we mark an inverted region between the HPRC1 and HPRC2 assemblies visible in the distal region of LCRD. ROI - region of interest (T2T-CHM13:18-23Mbp).

Supplementary Figure 49: Validation of singleton LCRA structures using HPRC2 (release 2) assemblies.

Singleton LCRA structures (n=47) observed among 133 noninverted haplotypes are presented in each row (always on the top). Then, for each singleton haplotype, we present the closest haplotype structure observed among HPRC2 assemblies (always at the bottom). Each haplotype structure is presented as directional arrowheads colored by a SD repeat unit (120 kbp, 105 kbp, 25 kbp, and 28 kbp; see legend; 41 kbp repeat unit not shown for simplicity). Singleton haplotypes whose exact structure was observed among HPRC2 samples are marked by green points.

Supplementary Figure 50: Representation of LCRA assemblies according to fiber-FISH duplcons.

Each haplotype has been decomposed in duplcons based on its fiber-FISH probe composition. The yellow, cyan, magenta, blue, green and white duplcons are 25 kbp, 29 kbp, 155 kbp, 100 kbp, 80 kbp and 31 kbp, respectively (Demaerel et al. 2019). LCRA structures defined by fiber-FISH duplcons and assembly are very similar in content, size, and orientation. The corresponding assembly haplogroup is given on the right. The assembly-based SD repeat units, defined in this manuscript, blue (25 kbp), yellow (28 kbp), light red (105 kbp) and brown (120 kbp) match the white (31 kbp), cyan (29 kbp), blue (100 kbp) and magenta (155 kbp) (see the legend). The fiber-FISH-based duplcon with blue stripes matches the 41 kbp duplcon defined within our assemblies (Fig. 2c).

Supplementary Figure 51: Validation of 22q11.2DS families using fiber-FISH.

LCRA and LCRD alleles of proband and parent of origin are represented using the SD repeat units as published by Demareel et al. (2019). The region in which the breakpoint occurred is shown for each duo in a black box. The assembled haplotypes have been decomposed in duplicons based on the fiber-FISH probe composition. All haplotypes matched the fiber-FISH defined structure.

3. Summarization of results could be improved

The main text sometimes moves quickly between specific findings without offering overarching summaries. For example, the description of inversion susceptibility and differences between populations (e.g., between African and European haplotypes) is fascinating but could be better synthesized.

A summary figure or model illustrating which structural features (e.g., SD orientation, duplication content) predispose to deletions vs inversions would make the core results more accessible to a broad readership.

We reworked the text to include additional details before summarizing main conclusions. Based on a suggestion from Reviewer #1, we also performed additional (and more relevant) predisposition analyses relevant to differences in ancestry. In addition to supporting the original observation of African ancestry protection, we now find that East Asians possess sequence configurations that would make them more susceptible to 22q11.2 deletions. Finally, we revised our conclusions based on suggestions from all reviewers (see updated Discussion below). To better illustrate the mechanism of NAHR and its possible outcomes given the relative orientation of paralogous segmental duplications, we added a new panel to the **Extended Data Figure 1** (see below panel a).

Extended Data Figure 1: Predisposition to 22q11.2DS via intrachromosomal NAHR.

a) Model of intra- and interchromosomal non-allelic homologous recombination (NAHR) and its possible outcomes depending on the relative orientation of segmental duplications (or low copy repeats). **b**) Counts of all possible LCRA and D interactions of the 105 kbp repeat unit per single haplotype (n=134, including T2T-CHM13 haplotype). Only intrachromosomal (cis) interactions are evaluated. Direct (forward)-oriented pairs are colored in dark red and inverted (reverse) pairs are shown in gray. At the base of the plot we show the ancestry origin of each haplotype. **c**) Shows evaluation of the proportions of each ancestry (AFR - African, AMR - American, EUR - European, EAS - East Asian, and SAS - South-East Asian) among haplotypes defined as either predisposed or protected against 22q11.2DS with respect to the rest of the ancestries ('Other' - gray). We used Fisher's exact test (two-sided) to test the significance of the difference between these proportions. Above each barplot is a reported p-value after Bonferroni correction.

Revised discussion points:

Importantly, these expansions do not always correlate with a predicted increased probability for NAHR. Considering percent identity and repeat orientation between LCRA and D, an analysis of both interchromosomal and intrachromosomal configurations predicts that individuals of African

ancestry are more likely to be predisposed to inversions as opposed to large-scale *de novo* deletions (**Extended Data Fig. 1c, Fig. 5c**). Thus, despite the larger size of LCRA and higher copy number of the 105 kbp SD repeat unit, Africans are more likely to be fully protected against 22q11.2DS than those of non-African ancestry because of the preponderance of inversely oriented repeats between the LCR blocks. This sequence structure prediction matches closely an epidemiological observation made 20 years ago where 22q11.2DS deletions were found to be three times less common among patients of African-American ancestry (McDonald-McGinn et al. 2005; Pastor et al. 2025). This reduced prevalence was subsequently shown not to be the result of phenotypic ascertainment biases and was hypothesized to be the result of differences in the LCR structures (McDonald-McGinn et al. 2023). In contrast, we find that the LCR structures of individuals of East Asian ancestry are significantly (Fisher's exact test, two-sided, Bonferroni corrected, $p = 1.19 \times 10^{-7}$, odds ratio = 0) predisposed to interchromosomal NAHR due their predominant direct orientation of the 105 kbp repeat in between LCRA and D; however, we were not able to find clinical results to support this observation. Nevertheless, these findings suggest that it might be more important to screen individuals of East Asian descent (i.e., Japan) for 22q11.2DS given their particular structural configurations that predispose to NAHR (**Supplementary Fig. 46**).

4. Functional or clinical implications not fully developed

While the structural and evolutionary findings are strong, the potential clinical or mechanistic relevance could be emphasized further.

For example, how might these findings inform carrier screening, CNV risk stratification, or population-specific variant interpretation?

Would these structures affect mapping or CNV-calling in short-read data?

To address this question, we updated our deletion predisposition at 22q11.2. To increase the number of samples with both haplotypes assembled we also included assemblies from HPRC2 (release 2) to arrive at 233 samples to test in total. For each sample we tested the sequence identity and relative orientation of interchromosomal interaction of the 105 kbp repeat unit between LCRA and D. We marked samples with the majority ($\geq 75\%$) of all possible interactions between LCRA and D in forward orientation as predisposed while samples with the majority ($\geq 75\%$) of these interactions in reverse orientation are marked as protected. We observed that samples of African ancestry are significantly (Fisher's exact test, two-sided, Bonferroni corrected, $p = 1.14 \times 10^{-6}$, odds ratio = 25.8) more protected against 22q11.2DS while samples of East Asian ancestry tend to be more predisposed (Fisher's exact test, two-sided, Bonferroni corrected, $p = 1.19 \times 10^{-7}$, odds ratio = 0) (**Fig. 5c**). Other ancestries do not show such strong biases towards predisposition or protection against 22q11.2DS. Thus, these findings would suggest that it will be more important to screen individuals of East Asian descent (i.e., Japan) as our results suggest that 22q11.2DS would be more prevalent (**Supplementary Fig. 46**).

Figure 5: 22q11.2DS predisposition and microdeletion rearrangements.

a) A schematic showing intrachromosomal (cis) and interchromosomal (trans) unequal crossover between the 105 kbp repeat unit within LCRA and LCRD. Forward- and reverse-oriented 105 kbp repeat units between LCRA and D are shown in dark red and gray, respectively. **b**) Counts of all possible LCRA and D interactions of the 105 kbp repeat unit per sample ($n=233$, plus four 22q11.2DS transmitting parents). Only interchromosomal (trans) interactions are evaluated (see Extended Data Fig. 1 for intrachromosomal interactions). Direct (forward)-oriented pairs are colored in dark red and inverted (reverse) pairs are shown in gray. At the base of the plot we show the ancestry origin of each haplotype. Black arrows mark the results for four transmitting parents. **c**) Evaluation of the proportions of each ancestry (AFR - African, AMR - American, EUR - European, EAS - East Asian, and SAS - South-East Asian) among haplotypes defined as either predisposed or protected against 22q11.2DS with respect to the rest of the ancestries ('Other' - gray). We used Fisher's exact test (two-sided) to test the significance of the difference between these proportions. Above each barplot is the reported p -value after Bonferroni correction. **d**) Visualization of binned (bin size: 10 kbp) sequence identity between probands with respect to both haplotypes (H1 and H2) of a transmitting parent for four families. The proband haplotype is in the middle while parental haplotypes H1 and H2 are at the top and bottom, respectively. Alignments are colored by sequence identity and likely inherited portions of parental haplotypes in the proband are highlighted by fully colored alignments (darker shades of red) while others are shown with 50% transparency. On top of each haplotype are HOR annotations depicted as arrowheads colored by SD repeat units. Refined deletion

breakpoints within the 105 kbp repeat units are marked by black arrowheads. **e)** Coverage summary of the longest continuous region of high identity ($\geq 99.5\%$) among all alignments between all possible interchromosomal interactions of 105 kbp long repeat units per sample. We distinguish the coverage for the direct (forward, dark red) and inverted (reverse, gray) alignments. We mark the position of putative deletion breakpoints in four probands as black arrowheads.

Supplementary Figure 46: Population labels of samples predicted to be 22q11.2DS susceptible.

The height of each bar shows the number of samples marked as predisposed to 22q11.2DS colored by population labels obtained from the 1KG project. The three letter abbreviations are 1KG-specific population labels (x-axis). Shades of green mark individuals of East-Asian ancestry (EAS).

To answer the second part of the question as to how complex duplication structures at 22q11.2 affect CNV prediction using short-read data, we performed a copy number analysis of 75 samples (out of 78) presented in this study along with seven samples with 22q11.2DS (Cleyne et al. 2021). While short-read data can detect samples that carry large-scale deletions between LCRA and D and LCRA and B, such data are unable to pinpoint exact SD repeat units involved in breakpoint formation (Supplementary Fig. 1). This is because short-read sequencing (150-250 bp Illumina reads) is unable to distinguish the copy number of SDs located at the breakpoints as unequal crossover typically occurs in stretches of paralogy of near-perfect sequence identity. Thus, LRS is necessary to reconstruct the structures identifying predisposed and protective configurations for the 22q11 deletion; however, it is not crucial to detect a deletion or duplication of the critical region, since such regions would be readily assayed by short-read sequence read depth (Supplementary Fig. 1). We added a notion of this in the discussion (see below).

Discussion point revised (in bold):

The structural complexity of the LCRA-D duplication blocks at the 22q11.2 locus and the resulting challenges in accurately and completely assembling this region have limited our ability to fully characterize the mechanisms driving these rearrangements. **For instance, short-read sequencing methods are optimal for detecting large-scale deletions at 22q11.2; however, they are unable to distinguish the copy number of SDs located at the deletion breakpoints as unequal crossover typically occurs in stretches of paralogy with near-perfect sequence identity**(Cleyne et al. 2021) (Supplementary Fig. 1). Other studies using

fluorescence *in situ* hybridization (FISH) and optical mapping(Shimojima et al. 2011; Vervoort and Vermeesch 2022) provided valuable insights into the structural diversity of this locus, but these methods lack the resolution necessary to precisely map deletion breakpoints or explore the fine-scale diversity of the underlying LCRs flanking this region. More recently, optical mapping methods were coupled with ultra-long Oxford Nanopore Technologies (UL-ONT) sequencing approaches to locally assemble and characterize breakpoints in 22q11.2DS families(Vervoort et al. 2025; Zhou et al. 2024).

Supplementary Figure 1: Copy number profiles for 75 samples reported in this paper’s human diversity panel.

A copy number (CN) heatmap based on high-coverage Illumina read-depth profiles across the 22q11.2 region (T2T-CHM13 coordinates; chr22:18000000-23000000) plotted separately for the 75 samples from the 1KG panel presented in this study. In addition, there are seven samples with 22q11.2DS. Of those, there are five samples with LCRA-D and two with LCRA-B deletions (one sample per row). Color designates the estimated copy number (see legend). The left colored bar highlights samples of the same ancestry and 22q11.2DS samples. At the top there are annotations of common repeat units (25, 28, 105, 120, and 41 kbp in size) as well as DupMasker annotation. We also mark positions of LCRA and LCRD where the highest copy number (≥ 10) is reported, which prevents accurate estimates of copies of each common repeat unit.

Minor Points

5. Use of color and labels in figures

Several figures use similar colors to distinguish critical structural features, making it hard to distinguish them in print or grayscale.

Suggestion: Use more distinct color palettes and add direct in-figure labeling or legends (rather than relying on external figure captions).

We modified all our figures accordingly to address this concern. Specifically, we made an effort to increase the font in all main and supplementary figures. We also improved color palette discernment whenever possible.

Example Figure 1 with all labels increased in size.

Reviewer #3 (Remarks to the Author):

Porubsky et al. describe population diversity and evolution and of four complex and unstable segmental duplication clusters (LCRA-D) in the 5 Mb Chromosome 22q11.2 deletion syndrome region. Using computational analyzes, including de novo genome assembly of long read PacBio HiFi and ONT whole genome sequencing and low-coverage Strand-seq data, they sequence-resolved 135 complete human 22q11.2 haplotypes from diverse ethnicities (African, American, East Asian, Southeast Asian, and European, 1000 Genome Project) and syntenic regions from six nonhuman primate species. Consistent with previous studies, LCRA was found to be most variable. They defined a 105 kb higher-order SD cassette flanked by 25 kb inverted repeats adjacent to PATRR sequences, as a major genomic instability driver that had emerged in the human-chimpanzee ancestral lineage and has expanded in humans less than 1.0 million years ago. They found that LCRA-D deletion breakpoints map to the 105 kb repeat unit whereas large rare inversion and inverted duplication breakpoints map to the 25 kb repeats. Corroborating previous clinical observations, African LCR-A haplotypes were found to be protecting against recurrent microdeletions, but more predisposing to genomic inversions because of the preponderance of inversely oriented repeats.

The manuscript is well-written and adds to the literature.

We thank the reviewer for recognizing the value of this study and for all their useful comments.

1) My major criticism is that the Authors present little molecular validation of their computational findings.

Since all three reviewers had a similar request, we created a new Supplementary Note (see below) for the manuscript to summarize all of our assembly quality filtering steps.

With respect to molecular validation, we used previously published fiber-FISH datasets to validate applicable haplotypes from our human diversity panel (n=133). Additionally, the assembly of the patients and family members was confirmed using fiber-FISH.

Supplementary Note 1:

Evaluation of 22q11.2 assembly accuracy of human diversity panel. To ensure the quality of assembled haplotypes analyzed in this study, we set to evaluate the assembly accuracy using multiple orthogonal methods and datasets.

- 1) From the total of 220 haplotypes, we selected only those haplotypes with a region corresponding to the 22q11.2 region (chr22:18000000-23000000, T2T-CHM13 coordinates) assembled in a single continuous contig with no N's within a sequence that would point to scaffolded assembly gaps.
- 2) We re-aligned PacBio HiFi reads back to each assembly using minimap2 (**Methods**) in order to evaluate each haplotype assembly for the presence of assembly collapses in highly identical LCR regions (especially in LCRA and D) (Vollger et al. 2019). The

presence of an increased coverage over a defined region along with an increased count of the second most abundant allele is indicative of a collapsed assembly of a piece of DNA present in a given region in more than one copy (**Supplementary Fig. 47, Supplementary Table 8**).

- 3) To further evaluate the accuracy of HPRC (release 1) assemblies generated using HiFi reads only (Liao et al. 2023), we obtained access to HPRC2 (release 2) assemblies completed using both HiFi and ONT reads. For a subset of these samples, we were able to compare assembly accuracy for 35 haplotypes. Of the 35 haplotypes, there are 30 for which the sequence identity between release 1 and release 2 HPRC assemblies is more than 99.99% (**Supplementary Fig. 48**). We noticed, however, five haplotypes with a lower sequence accuracy (>99.5% but <99.99%) due to an inversion in a distal part of the LCRD. We predict this is a misassembly within HPRC (release 1) assemblies, so we now report affected haplotypes in **Supplementary Table 8**. We do still use these five haplotypes in our subsequent analysis as these low-frequency discrepancies have no effect on our prediction of deletion predisposition at 22q11.2 because of the location of the putative misassembly.
- 4) As another metric of accuracy, we searched for biological replication—i.e., evidence that the same structure had been observed in two independent human genomes. Specifically, we took advantage of available HPRC2 (release 2) data (**Data Availability**) to evaluate the accuracy of the higher-order structure of LCRA among all singleton haplotypes (n=47) defined among a total of 63 unique LCRA structures. For each singleton structure we searched the 418 HPRC2 structures of LCRA in order to find the best matching one. We did this by encoding each repeat unit by a unique number and comparing them against all HPRC2 structures. We calculated the distance between these numeric encodings of each structure using R package stringdist and its function 'seq_dist' (function parameter: method='osa'). We consider a singleton structure to be validated by HPRC2 if there is a matching HPRC2 structure with less ≤5% divergence. Based on this threshold, we confirm 23 singleton structures are seen at least once in the HPRC2 dataset (**Supplementary Fig. 49**).
- 5) Lastly, we compared the organization of LCRA to previous fiber-FISH and optical-mapping-based analyses of the same samples where applicable. To validate the assemblies, the structure of LCRA among the 133 haplotypes has been investigated using the sequence of the probes previously used to characterize the region by fiber-FISH and optical genome mapping (Demaerel et al. 2019). The probe content of each haplotype was inferred using BLAST and selecting matches with at least 87% identity compared to the fiber-FISH probe and with length of at least 50% of the fiber-FISH probe. LCRAs are represented using the repeat units defined by Demaerel et al. (2019). Most assemblies have a structure that matches what has been previously described. Haplogroups 13* and 14 differ slightly from previously described structures as one probe is only partially included (42% of the probe length is included in the assembly) (**Supplementary Fig. 50**). A detailed comparison of the structures is summarized in **Supplementary Table 9**. Similarly, we also validated the assembly-based structure of LCRA and D using fiber-FISH for three family duos (AD009, AD010, and AD013) (Demaerel et al. 2019). All family duos are in line with the fiber-FISH observed LCRA

and D structure (**Supplementary Fig. 51**). The structural validity of the Coriell trio was confirmed using local reassembly with NOVOloci (**Methods**). We, however, observe a putative haplotype switch error downstream from LCRD, which has no effect on the proper mapping of the 22q11.2DS breakpoint (**Supplementary Fig. 44**).

We note that the chromosome-scale phasing accuracy of all our assembled haplotypes was ensured by using trio-based phasing in HPRC (release 1) (Liao et al. 2023) assemblies or using Strand-seq phasing information in HG SVC (Logsdon et al. 2025) assemblies.

Supplementary Figure 47: Phased assembly validation using NucFreq.

Two examples of read-coverage profiles of aligned PacBio HiFi reads to the two phased genome assemblies of the 22q11.2 region (chr22:1800000-23000000 in T2T-CHM13 coordinates). **a**) A visualization of the most abundant base at each position of HG04036-H2 is shown in black. The second most abundant base is shown in red. Extended regions with observable increased coverage for the most abundant base ('Max') often coincide with increased frequency of the second ('Remainder') most abundant base (see arrow). Such regions are indicative of a collapsed assembly where not all copies of paralogous duplications are fully resolved. Misassemblies would be represented as an absence of sequence reads mapping across a region. **b**) This visualization shows the same frequency of the most and second most abundant bases along the haplotype from sample NA19705-H1. In this case we consider this assembly to be complete and correct as there are no extended regions of increased frequency of the secondary base ('Remainder' base).

Supplementary Figure 48: Comparison of HPRC1 and HPRC2 assemblies for the same haplotypes.

a) A barplot showing the percent agreement between HPRC1 (release 1) and HPRC2 (release 2) assemblies generated using PacBio HiFi reads only for HPRC1 and both HiFi and ONT reads for HPRC2 assemblies. Haplotypes marked with an asterisk are those where we observed a distal inversion within LCRD (see example in panel b). **b)** Visualization of binned (bin size: 10 kbp) sequence identity between the HPRC2 assembly for samples HG01071-H2 aligned to the HPRC1 assembly of the same haplotype. Alignments are colored by sequence identity. On top of the HPRC1 assembly there is a DupMasker annotation shown as colored arrowheads. Continuous LCRA and D regions are also highlighted. There is also a structural variant (SV, ≥ 50 bp) colored by blue and red outlines for insertions (INS) and deletions (DEL), respectively. Lastly, we mark an inverted region between the HPRC1 and HPRC2 assemblies visible in the distal region of LCRD. ROI - region of interest (T2T-CHM13:18-23Mbp).

Supplementary Figure 49: Validation of singleton LCRA structures using HPRC2 (release 2) assemblies.

Singleton LCRA structures (n=47) observed among 133 noninverted haplotypes are presented in each row (always on the top). Then, for each singleton haplotype, we present the closest haplotype structure observed among HPRC2 assemblies (always at the bottom). Each haplotype structure is presented as directional arrowheads colored by a SD repeat unit (120 kbp, 105 kbp, 25 kbp, and 28 kbp; see legend; 41 kbp repeat unit not shown for simplicity). Singleton haplotypes whose exact structure was observed among HPRC2 samples are marked by green points.

Supplementary Figure 50: Representation of LCRA assemblies according to fiber-FISH duplcons.

Each haplotype has been decomposed in duplcons based on its fiber-FISH probe composition. The yellow, cyan, magenta, blue, green and white duplcons are 25 kbp, 29 kbp, 155 kbp, 100 kbp, 80 kbp and 31 kbp, respectively (Demaerel et al. 2019). LCRA structures defined by fiber-FISH duplcons and assembly are very similar in content, size, and orientation. The corresponding assembly haplogroup is given on the right. The assembly-based SD repeat units, defined in this manuscript, blue (25 kbp), yellow (28 kbp), light red (105 kbp) and brown (120 kbp) match the white (31 kbp), cyan (29 kbp), blue (100 kbp) and magenta (155 kbp) (see the legend). The fiber-FISH-based duplcon with blue stripes matches the 41 kbp duplcon defined within our assemblies (Fig. 2c).

Supplementary Figure 51: Validation of 22q11.2DS families using fiber-FISH.

LCRA and LCRD alleles of proband and parent of origin are represented using the SD repeat units as published by Demareel et al. (2019). The region in which the breakpoint occurred is shown for each duo in a black box. The assembled haplotypes have been decomposed in duplicons based on the fiber-FISH probe composition. All haplotypes matched the fiber-FISH defined structure.

2) Introduction

This deletion, also known as DiGeorge or velocardiofacial syndrome
 Please rephrase - a deletion is not a syndrome.

Revised (in bold):

In ~85% of cases, NAHR occurs between LCRA and D resulting in a deletion of a ~3 Mbp long critical region. This deletion **is the cause of** DiGeorge or velocardiofacial syndrome, which is now collectively referred to as the chromosome 22q11.2 deletion syndrome (22q11.2DS).

3) Page 4, Results

We initially evaluated the completeness of 220 phased genome assemblies generated as part of the HPRC18 and HGSVC19. Of these, 135 haplotypes were unrelated and assembled in a single continuous contig, passing our assembly quality validations²²

It is unclear to what extent these 135 haplotypes were assembled in the previous works PMID: 37165242 and 40702183. What do they mean by “evaluation of the completeness”?

All the initial assemblies from the 1000 Genomes population cohort were produced as part of the Human Genome Structural Variation Consortium (HGSVC) where our lab generated a large portion of long-read datasets as well as contributed to genome assembly work (Logsdon et al. 2025). Long-read sequence datasets (PacBio HiFi and UL-ONT) from the four 22q11.2 deletion families (three parent-child duos and one family trio) were generated as part of this publication. In addition, we generated long-read datasets (PacBio HiFi and UL-ONT) for two samples that carry the LCRA-D inversion (HG01888 and NA19315) along with Strand-seq data for predicted carriers (n=13) of the LCRA-D inversion found in HG01888. We also generated single-cell

Strand-seq datasets for an additional 279 samples to boost the prediction value of population frequencies for large-scale inversions at 22q11.2. Finally, our expanded analysis using HPRC2 (release 2) assemblies was recently released and our lab played a role in generating approximately one-third of that HiFi sequence data.

We have ensured that all these data sources are properly summarized in the **Data Availability** section.

When talking about assembly completeness evaluation, we mean first making sure that the donor specific assembly (DSA) traverses the 22q11.2 region in a single continuous contig and then we checked if PacBio HiFi reads mapped back to each DSA to support this assembly such that there are no obvious sequence collapses or assembly misjoins. Per the request from other reviewers, we now summarize all assembly quality steps in the **Supplementary Note 1** (see above).

4) Page 5, Results

We assigned SDs to LCR regions...

Do they distinguish SDs from LCRs? In the Abstract, they state “segmental duplications (SDs) known as low copy repeats (LCRs).

The two terms are synonymous and we did not mean to make a distinction between SD and LCRs. Those are the two different terms used in the literature to talk about repeats that are ≥ 1 kbp in size and $>95\%$ identical.

Revised (in bold):

The most common genomic disorder, chromosome 22q11.2 microdeletion syndrome (22q11.2DS), is mediated by highly identical and polymorphic segmental duplications (SDs) **also** known as low copy repeats (LCRs; regions A-D) that have been challenging to sequence and characterize. **We use both terms interchangeably throughout the manuscript.**

5) Page 7, Results, para 2, lines 5-6

Illegible statement.

We apologize for this though believe it may have been a system conversion problem as we cannot locate this issue, even though called out by two reviewers. However, we have carefully checked our manuscript for resubmission.

6) Page 17, Discussion

We find that inversion breakpoints cluster near PATRRs associated with the 25 kbp repeat unit
What do they mean by near? Please be more precise.

We rephrased our statements to remove this ambiguity and state only that the inversion breakpoints map to the 25 kbp repeat unit based on the observation in Figure 4d.

7) References 11, 15, 16, 17, 19, 36, 56, 59, 61 are incomplete or incorrect.

In our revised manuscript we checked the correct formatting of all citations. Thank you for catching these inconsistencies.

8) The manuscript is not double-spaced.

We submit our revised manuscript in double-spaced format for ease of review.

REFERENCES:

- Cleynen I, Engchuan W, Hestand MS, Heung T, Holleman AM, Johnston HR, Monfeuga T, McDonald-McGinn DM, Gur RE, Morrow BE, et al. 2021. Genetic contributors to risk of schizophrenia in the presence of a 22q11.2 deletion. *Mol Psychiatry* **26**: 4496–4510.
- Cooper GM, Coe BP, Girirajan S, Rosenfeld JA, Vu TH, Baker C, Williams C, Stalker H, Hamid R, Hannig V, et al. 2011. A copy number variation morbidity map of developmental delay. *Nat Genet* **43**: 838–846.
- Correll-Tash S, Lilley B, Salmons H Iv, Mlynarski E, Franconi CP, McNamara M, Woodbury C, Easley CA, Emanuel BS. 2021. Double strand breaks (DSBs) as indicators of genomic instability in PATRR-mediated translocations. *Hum Mol Genet* **29**: 3872–3881.
- Demaerel W, Mostovoy Y, Yilmaz F, Vervoort L, Pastor S, Hestand MS, Swillen A, Vergaelen E, Geiger EA, Coughlin CR, et al. 2019. The 22q11 low copy repeats are characterized by unprecedented size and structural variability. *Genome Res* **29**: 1389–1401.
- Dierckxsens N, Mansfield MJ, Plessy C, Luscombe NM, Vermeesch JR. 2025. Unlocking the full potential of Oxford Nanopore reads with NOVOloci. *bioRxiv*. <http://dx.doi.org/10.1101/2025.08.08.669243>.
- Guo X, Delio M, Haque N, Castellanos R, Hestand MS, Vermeesch JR, Morrow BE, Zheng D. 2016. Variant discovery and breakpoint region prediction for studying the human 22q11.2 deletion using BAC clone and whole genome sequencing analysis. *Hum Mol Genet* **25**: 3754–3767.
- Inoue K, Lupski JR. 2002. Molecular mechanisms for genomic disorders. *Annu Rev Genomics Hum Genet* **3**: 199–242.
- Kato T, Kurahashi H, Emanuel BS. 2012. Chromosomal translocations and palindromic AT-rich repeats. *Curr Opin Genet Dev* **22**: 221–228.
- Kurahashi H, Inagaki H, Hosoba E, Kato T, Ohye T, Kogo H, Emanuel BS. 2007. Molecular cloning of a translocation breakpoint hotspot in 22q11. *Genome Res* **17**: 461–469.
- Liao W-W, Asri M, Ebler J, Doerr D, Haukness M, Hickey G, Lu S, Lucas JK, Monlong J, Abel HJ, et al. 2023. A draft human pangenome reference. *Nature* **617**: 312–324.
- Logsdon GA, Ebert P, Audano PA, Loftus M, Porubsky D, Ebler J, Yilmaz F, Hallast P, Prodanov T, Yoo D, et al. 2025. Complex genetic variation in nearly complete human genomes. *Nature* **644**: 430–441.
- McDonald-McGinn D, Crowley T, McGinn D, Gaiser K, Giunta V, Lairson L, Zackai E, Tran O, Share M, Valverde K, et al. 2023. P248: Is 22q11.2 deletion syndrome truly less common in Black patients? *Genet Med Open* **1**: 100276.
- McDonald-McGinn DM, Minugh-Purvis N, Kirschner RE, Jawad A, Tonnesen MK, Catanzaro JR, Goldmuntz E, Driscoll D, Larossa D, Emanuel BS, et al. 2005. The 22q11.2 deletion in African-American patients: an underdiagnosed population? *Am J Med Genet A* **134**: 242–246.

- Pastor S, Tran O, Jin A, Carrado D, Silva BA, Uppuluri L, Abid HZ, Young E, Crowley TB, Bailey AG, et al. 2020. Optical mapping of the 22q11.2DS region reveals complex repeat structures and preferred locations for non-allelic homologous recombination (NAHR). *Sci Rep* **10**: 12235.
- Pastor S, Tran O, Lapointe R, Olali AZ, Wallace DC, Morrow B, Zackai EH, McDonald-McGinn DM, Emanuel BS. 2025. Optical mapping in Black genomes: Distinct LCR22 structures and 22q11.2 deletion syndrome mechanisms. *Genet Med* **28**: 101614.
- Porubsky D, Dashnow H, Sasani TA, Logsdon GA, Hallast P, Noyes MD, Kronenberg ZN, Mokveld T, Koundinya N, Nolan C, et al. 2025. Human de novo mutation rates from a four-generation pedigree reference. *Nature*. <http://dx.doi.org/10.1038/s41586-025-08922-2>.
- Saitta SC, Harris SE, Gaeth AP, Driscoll DA, McDonald-McGinn DM, Maisenbacher MK, Yersak JM, Chakraborty PK, Hacker AM, Zackai EH, et al. 2004. Aberrant interchromosomal exchanges are the predominant cause of the 22q11.2 deletion. *Hum Mol Genet* **13**: 417–428.
- Shimajima K, Okamoto N, Inazu T, Yamamoto T. 2011. Tandem configurations of variably duplicated segments of 22q11.2 confirmed by fiber-FISH analysis. *J Hum Genet* **56**: 810–812.
- Vervoort L, Dierckxsens N, Sousa Santos M, Meynants S, Souche E, Cools R, Heung T, Devriendt K, Peeters H, McDonald-McGinn DM, et al. 2025. Multiple paralogs and recombination mechanisms contribute to the high incidence of 22q11.2 deletion syndrome. *Genome Res* **35**: 786–797.
- Vervoort L, Vermeesch JR. 2022. The 22q11.2 low copy repeats. *Genes (Basel)* **13**: 2101.
- Vollger MR, Dishuck PC, Sorensen M, Welch AE, Dang V, Dougherty ML, Graves-Lindsay TA, Wilson RK, Chaisson MJP, Eichler EE. 2019. Long-read sequence and assembly of segmental duplications. *Nat Methods* **16**: 88–94.
- Zhou B, Purmann C, Guo H, Shin G, Huang Y, Pattni R, Meng Q, Greer SU, Roychowdhury T, Wood RN, et al. 2024. Resolving the 22q11.2 deletion using CTLR-Seq reveals chromosomal rearrangement mechanisms and individual variance in breakpoints. *Proc Natl Acad Sci U S A* **121**: e2322834121.